# Topological superconducting vortex from trivial electronic bands

Lun-Hui Hu [1,2] & Rui-Xing Zhang [1,2,3] ✉

Superconducting vortices are promising traps to confine non-Abelian Majorana quasi-particles. It has been widely believed that bulk-state topology, of either normal-state or superconducting ground-state wavefunctions, is crucial for enabling Majorana zero modes in solid-state systems. This common belief has shaped two major search directions for Majorana modes, in either intrinsic topological superconductors or trivially superconducting topological materials. Here we show that Majorana-carrying superconducting vortex is not exclusive to bulk-state topology, but can arise from topologically trivial quantum materials as well. We predict that the trivial bands in superconducting HgTe-class materials are responsible for inducing anomalous vortex topological physics that goes beyond any existing theoretical paradigms. A feasible scheme of strain-controlled Majorana engineering and experimental signatures for vortex Majorana modes are also discussed. Our work provides new guidelines for vortex-based Majorana search in general superconductors.

In condensed matter systems, the marriage of topology and electron correlations allows for fractionalizing electronic degrees of freedom into exotic non-Abelian quasiparticles such as Majorana zero modes (MZMs)[1,2]. Research efforts in the past two decades have together established superconductors (SCs) with certain topological properties as the best venue for trapping and manipulating MZMs, with which quantum information can be processed in a topologically protected manner. For example, a topological SC (TSC) can host zero-dimensional (0D) MZMs bound to either its geometric boundary[3] or the superconducting vortex[4], a manifestation of the bulk-boundary correspondence principle. This scenario has motivated enormous research efforts in unconventional SCs and ferromagnet-SC heterostructures[5–8], where natural and artificial TSCs are believed to exist, respectively. Remarkably, such a topological requirement can be further relaxed for vortex-trapped MZMs if the bulk electronic band structure, instead of the superconductivity itself, carries a nontrivial topological index[9,10]. This spirit also inspires another intensive search of topological-band materials with intrinsic yet non-topological SC[11–18], with many promising candidates discovered[19–22]. However, as far as we know, the possibility of trapping MZMs in trivial s-wave SCs with trivial electronic band structures has been rarely explored in the literature.

In this work, we show that a three-dimensional (3D) s-wave spin-singlet SC, with certain non-topological normal states, is capable of harboring Majorana-carrying topological vortices. This conclusion is explicitly demonstrated in the superconducting phase of 3D Luttinger semimetal (LSM)[23] as a proof of concept, whose normal-state semi-metallicity is of trivial topology. Topological superconducting vortex-line states with either 0D end-localized MZMs or a 1D Dirac-nodal dispersion are found to be ubiquitous in the vortex phase diagram of LSMs, shedding new light on this 60-year-old classical band system. The vortex-line topology here manifests a distinct origin from known vortex Majorana theories[9–17,24–28], most of which would require topological band inversion in the normal states. Furthermore, a tensile-strained LSM is found to be a bulk-trivial yet vortex-exotic band insulator, which harbors distinct topological vortex phases in the presence of electron and hole dopings, respectively.

LSMs generally show up as the $\Gamma_8$ quartet in HgTe-class materials, where the inversion between $\Gamma_8$ and $\Gamma_6$ bands usually creates a zero-gap topological insulator (TI). The composition of TI and LSM bands offers a minimal exemplar to visualize the competition between topological and trivial bulk bands for deciding the vortex topology. While a topological-band-only analysis anticipates a Majorana-carrying Kitaev

[1]Department of Physics and Astronomy, The University of Tennessee, Knoxville, TN 37996, USA. [2]Institute for Advanced Materials and Manufacturing, The University of Tennessee, Knoxville, TN 37920, USA. [3]Department of Materials Science and Engineering, The University of Tennessee, Knoxville, TN 37996, USA. ✉e-mail: ruixing@utk.edu

vortex, our new vortex paradigm predicts a Majorana-free topological nodal vortex instead, further confirmed by our numerical simulations. We propose the lattice strain effect as a promising control knob to detect and engineer vortex MZMs in superconducting HgTe-class materials. Experimental signatures of the proposed vortex topological physics are discussed in detail. We conclude by highlighting the potentially crucial role of low-energy trivial bands in deciding the vortex topology in general SCs and further providing suggestions on the ongoing Majorana search.

## Results

### $C_n$-symmetric vortex topology

We start with a general topological discussion on the superconducting vortex-line states. A superconducting vortex in a 3D Bogoliubov-de Gennes (BdG) system is a 1D line defect that traps low-energy Caroli-de Gennes-Matricon (CdGM) bound states. Generated by an external magnetic field **B**, the CdGM states disperse along $\mathbf{k_B} \| \mathbf{B}$ to form an effective 1D system in symmetry class D, as described by a vortex-line Hamiltonian $h_{\text{vort}}(k_\mathbf{B})$. Throughout this work, we will denote $\hat{z}$ as the magnetic field direction for simplicity. Besides the built-in particle-hole symmetry (PHS), $h_{\text{vort}}$ can additionally respect $\mathcal{G}_\mathbf{B}$, a subgroup of the 3D crystalline group $\mathcal{G}$ in the zero-field limit. The band topology of $h_{\text{vort}}$ is protected by both PHS $\Xi$ and $\mathcal{G}_\mathbf{B}$.

We focus in this work on general s-wave spin-singlet superconductors, where $\mathcal{G}_\mathbf{B}$ is a n-fold rotation group $C_n$ and every CdGM state carries a $C_n$ index $J_z \in \{0, 1, 2, ..., n-1\}$, i.e., the $\hat{z}$-directional angular momentum modulo n. CdGM states with different $J_z$ labels are decoupled from each other along $k_z$ and each $J_z$ sector can be characterized by its own 1D topological index. With an s-wave pairing, $J_z \in \{0, \frac{n}{2}\}$ sectors are PHS invariant themselves and carry a $\mathbb{Z}_2$ Pfaffian index $\nu_{J_z} \in \{0,1\}$[3]. Note that for systems with a non-s-wave pairing, the PHS-invariant $J_z$ sectors might be different from the above. When $\nu_{J_z} = 1$, all $J_z$-indexed CdGM states constitute a 1D TSC phase that is equivalent to a Kitaev Majorana chain, contributing to a $J_z$-labeled vortex MZM on the sample surface. We dub this gapped vortex phase a Kitaev vortex. On the other hand, $J_z$ and $n-J_z$ form particle-hole conjugate sectors if $J_z \notin \{0, \frac{n}{2}\}$ and together carry a $\mathbb{Z}$-type topological index,

$$\mathcal{Q}_{J_z} = n_{J_z}^{(v)}(0) - n_{J_z}^{(v)}(\pi), \tag{1}$$

where $n_{J_z}^{(v)}(k_z)$ counts the number of $J_z$-carrying CdGM states with negative energy at $k_z$. A derivation of $\mathcal{Q}_{J_z}$ is provided in Supplementary Note 1. Physically, $\mathcal{Q}_{J_z}$ indicates the number of pairs of $C_n$-protected BdG nodal points along $k_z$, signaling a band-inverted gapless vortex state dubbed a nodal vortex. Kitaev and nodal vortices are elementary building blocks to construct general $C_n$-protected vortex topological phenomena.

We now demonstrate our classification scheme. For instance, the $C_2$ group possesses two PHS-invariant $J_z$ sectors $J_z = 0$ and $J_z = 1$, and a general $C_2$-invariant vortex can only harbor Kitaev vortices but not the nodal ones. The vortex topology is then characterized by $\nu_{0,1}$, thus being $\mathbb{Z}_2 \times \mathbb{Z}_2$ classified. When $\nu_0 = \nu_1 = 1$, a Majorana doublet emerges in the surface vortex core and the two MZMs will not mix for carrying distinct $J_z$ labels. Take $C_6$ as another example, the $(\mathbb{Z}_2)^2 = \mathbb{Z}_2 \times \mathbb{Z}_2$ part is contributed by the PHS-invariant sectors $J_z = 0$ and $J_z = 3$, similar to that in the $C_2$ case. In addition, $(J_z = 1, J_z = 5)$ and $(J_z = 2, J_z = 4)$ form two pairs of particle-hole conjugate sectors indicated by $\mathcal{Q}_1$ and $\mathcal{Q}_2$, so that only nodal vortices can occur in these sectors. This leads to another $\mathbb{Z} \times \mathbb{Z}$ contribution, promoting the classification of $C_6$-symmetric vortices to $(\mathbb{Z}_2)^2 \times (\mathbb{Z})^2$. We summarize the vortex topological classification and characterization for all $C_n$ groups in Table 1.

Notably, the protection of vortex-line topology is decided by both the bulk crystalline symmetry group and the magnetic field orientation. Thus, it is possible to realize distinct vortex topological

## Table 1 | Vortex topological classification of $C_n$-invariant s-wave spin-singlet superconductors

| Symmetry | $C_1$ | $C_2$ | $C_3$ | $C_4$ | $C_6$ |
|---|---|---|---|---|---|
| Classification | $\mathbb{Z}_2$ | $\mathbb{Z}_2 \times \mathbb{Z}_2$ | $\mathbb{Z}_2 \times \mathbb{Z}$ | $(\mathbb{Z}_2)^2 \times \mathbb{Z}$ | $(\mathbb{Z}_2)^2 \times (\mathbb{Z})^2$ |
| Invariant | $\nu_0$ | $\nu_{0,1}$ | $(\nu_0, \mathcal{Q}_1)$ | $(\nu_{0,2}, \mathcal{Q}_1)$ | $(\nu_{0,3}, \mathcal{Q}_{1,2})$ |

$\nu_{J_z} \in \mathbb{Z}_2$ is a symmetry-indexed topological invariant signaling the presence ($\nu_{J_z} = 1$) or absence ($\nu_{J_z} = 0$) of a $J_z$-labeled vortex Majorana zero mode (MZM). The $C_n$ topological charge $\mathcal{Q}_{J_z} \in \mathbb{Z}$ characterizes the symmetry-protected vortex band crossings (i.e., a nodal vortex) near zero energy. In principle, a vortex line is capable of carrying multiple 0D vortex MZMs and nodal bands that do not interfere with each other, as long as they are supported by distinct topological indices.

states in a single superconducting material by simply rotating the applied magnetic field. This clearly implies the absence of an exact one-to-one mapping between bulk-state and vortex-line topologies. This observation motivates us to explore the possibility of topological vortices inside a *completely trivial* SC, whose topological triviality manifests in both its Cooper-pair and normal-state wavefunctions.

### Vortex topology from trivial bulk bands

Our target trivial-band system is a 3D Luttinger semimetal (LSM), which is defined by a single fourfold degenerate quadratic band touching at $\Gamma$[23,29], i.e., the origin of the Brillouin zone (BZ). This band degeneracy arises from a 4D double-valued irreducible representation (irrep) $\Gamma_8$ of point groups such as $O$, $O_h$, and $T_d$. Unlike traditional topological semimetals[30–32], the point node of an LSM does not serve as a topological quantum critical point between two distinct lower-dimensional gapped topological phases, and is thus trivial in the topological sense. Remarkably, such a trivial band set, together with isotropic s-wave superconductivity, will give rise to nontrivial vortex topologies, which we will show below.

The $\Gamma_8$-bands are captured by the atomic basis $|\Psi_{\Gamma_8}\rangle = (|p_+, \uparrow\rangle, |p_+, \downarrow\rangle, |p_-, \uparrow\rangle, |p_-, \downarrow\rangle)^T$ with $\uparrow, \downarrow$ denoting the electron spin and $p_\pm = p_x \pm ip_y$ orbitals. Under this basis, we consider a $\mathbf{k} \cdot \mathbf{p}$ model Hamiltonian around $\Gamma$ that respects inversion, time-reversal, and around-$\hat{z}$-axis full rotation symmetries. In particular, $\mathcal{H}_{\text{LSM}} = \lambda_1 k^2 \gamma_0 + M(\mathbf{k})\gamma_5 + v_z k_z (k_x \gamma_{45} + k_y \gamma_{35}) - \sqrt{3}\lambda_2((k_x^2 - k_y^2)\gamma_{25} + 2k_x k_y \gamma_{15})$. Here, $M(\mathbf{k}) = m_1(k_x^2 + k_y^2) + m_2 k_z^2$ and the $4 \times 4$ $\gamma$-matrices are defined as $\gamma_1 = \sigma_x \otimes s_z$, $\gamma_2 = \sigma_y \otimes s_z$, $\gamma_3 = \sigma_0 \otimes s_x$, $\gamma_4 = \sigma_0 \otimes s_y$, $\gamma_5 = \sigma_z \otimes s_z$ with $\gamma_{mn} = -i\gamma_m \gamma_n$ and $\gamma_0 = \sigma_0 \otimes s_0$ the identity matrix. $\sigma$ and $s$ are Pauli matrices denoting the orbital and spin degrees of freedom, respectively. Without loss of generality, we set $\lambda_1 = 0$ in the following discussion, and the four bulk band dispersions are $E_\pm(\mathbf{k}) = \pm\sqrt{(m_1^2 + 3\lambda_2^2)k_\parallel^4 + (2m_1 m_2 + v_z^2)k_z^2 k_\parallel^2 + m_2^2 k_z^4}$ with $k_\parallel^2 = k_x^2 + k_y^2$. Therefore, $\mathcal{H}_{\text{LSM}}$ describes a quadratic semimetal with different in-plane and out-of-plane dispersions, serving as an anisotropic generalization of the conventional isotropic LSM model[23,29]. The isotropic limit can be achieved with $m_1 = -\frac{1}{2}m_2 = \lambda_2$ and $v_z = -2\sqrt{3}\lambda_2$, leading to $E_\pm(\mathbf{k}) = \pm 2|\lambda_2|k^2$ with $k^2 = k_\parallel^2 + k_z^2$. A dispersion plot for the isotropic LSM phase is shown in Fig. 1a. The superconductivity of LSMs is described by generalizing $\mathcal{H}_{\text{LSM}}$ into a BdG form,

$$\mathcal{H}_{\text{BdG}} = \begin{pmatrix} \mathcal{H}_{\text{LSM}}(\mathbf{k}) - \mu & \mathcal{H}_\Delta \\ \mathcal{H}_\Delta^\dagger & \mu - \mathcal{H}_{\text{LSM}}^*(-\mathbf{k}) \end{pmatrix}, \tag{2}$$

where $\mu$ is the chemical potential. $\mathcal{H}_\Delta = i\Delta(\mathbf{r})\gamma_{13}$ describes an isotropic s-wave spin-singlet pairing, making $\mathcal{H}_{\text{BdG}}$ carry a trivial bulk topology. A superconducting vortex line centering at $r = 0$ can be generated by $\Delta(\mathbf{r}) = \Delta_0 \tanh(r/\xi_0)e^{i\theta}$, with $(r, \theta)$ being the in-plane polar coordinates and $\xi_0$ the SC coherence length.

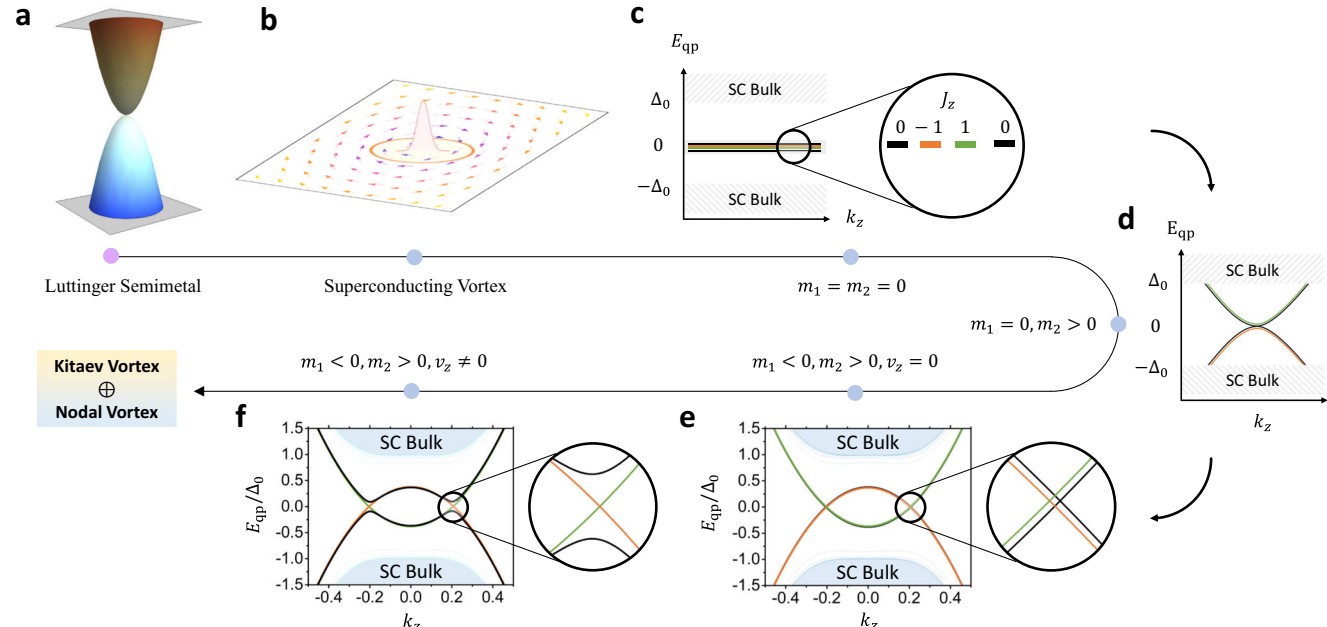

**Fig. 1 | Topological superconducting vortex in a Luttinger semimetal (LSM).** **a** shows the quadratic band touching around $\Gamma$ point of a LSM. In (**b**), the superconducting (SC) pairing function $\Delta(x, y)$ is illustrated for the $k_z = 0$ plane, where the vortex phase winding is denoted by in-plane arrows surrounding the vortex core. Four vortex zero modes are expected to occur for LSM at $k_z = 0$ due to an emergent chiral winding number. The vortex-line low-energy spectra $E_{qp}/\Delta_0$ are illustrated in (**c**) for $m_1 = m_2 = v_z = 0$ with four zero-energy flat bands labeled by angular momenta $J_z$; and in (**d**) for $m_1 = 0, m_2 > 0, v_z = 0$. Two pairs of vortex nodal bands show up in (**e**) for $v_z = 0$, while only the ones formed by $J_z = \pm 1$ are symmetry protected. Turning on $v_z \neq 0$ will gap out the unprotected crossings within $J_z = 0$ sector, as shown in (**f**), leading to a Kitaev vortex. The final vortex state of an LSM consists of a nodal vortex coexisting with a Kitaev vortex. e, f are numerically simulated in a disk geometry with band parameters $m_1 = -1, m_2 = 2, v_\parallel = \sqrt{3}, v_z = 2\sqrt{3}, \Delta_0 = 0.2$.

The origin of topological vortex-line modes in LSMs can be understood in a perturbative manner, which is schematically depicted in Fig. 1. This is motivated by a key observation that the normal state $\mathcal{H}_{\text{LSM}}(\mathbf{k}) = h^{(0)}(\mathbf{k}_\parallel) + h^{(1)}(\mathbf{k}_\parallel, k_z)$ with

$$h^{(0)}(\mathbf{k}_\parallel) = \begin{pmatrix} 0 & -\sqrt{3}\lambda_2 k_-^2 \\ -\sqrt{3}\lambda_2 k_+^2 & 0 \end{pmatrix} \otimes s_0. \quad (3)$$

Here $k_\pm = k_x \pm ik_y$. The unperturbed part $h^{(0)}$ describes two identical copies of 2D massless quadratic Dirac fermions, each of which carries a $2\pi$ Berry phase and is similar to those live in bilayer graphene[33] and on the surfaces of topological crystalline insulators[34,35]. While a 2D linear Dirac fermion carries a single vortex MZM[9], we naturally expect $h^{(0)}$ to support four vortex MZMs if going superconducting, with each quadratic Dirac fermion contributing a pair of MZMs in Fig. 1c.

This conjecture is confirmed by exactly mapping the 2D vortex problem of $h^{(0)}(\mathbf{k}_\parallel)$ to a 3D chiral topological insulator[36], thanks to an emergent chiral symmetry $\mathcal{S}$ of the system. This allows us to exploit the 3D chiral winding number $\mathcal{N}_\mathcal{S}$[37] to topologically quantify the zero modes, with the spatial polar angle $\theta$ acting as an extra dimension in addition to $k_x$ and $k_y$. As discussed in Methods, we analytically calculate $\mathcal{N}_\mathcal{S} = 4$, confirming these four vortex zero modes. We further simulate the superconducting vortex of $h^{(0)}(\mathbf{k}_\parallel)$ on a large disc geometry to numerically confirm the zero modes, and find that they are $J_z$-labeled. In particular, two zero modes form a PHS-related pair and carry $J_z = \pm 1$, while the other two are both labeled by $J_z = 0$.

Taking into account $h^{(1)}(\mathbf{k}_\parallel, k_z)$, the four zero modes start to hybridize, split, and disperse along $k_z$. Crucially, we note that in $h^{(1)}, M(\mathbf{k}) = m_1(k_x^2 + k_y^2) + m_2 k_z^2$ features $m_1 m_2 = -2\lambda_2^2 < 0$ for an isotropic LSM. As we rigorously prove in Supplementary Note 3, a negative $m_1$ will send two zero modes with $J_z = 0, 1$ [i.e. colored in black and green in Fig. 1c] to negative energy. Meanwhile, a positive $m_2$ will make sure the same zero modes to quadratically disperse

along $k_z$, but with a positive mass. The PHS requires the other two zero modes with $J_z = 0, -1$ to behave oppositely. As a result, the original quartet of zero modes evolves into two pairs of 1D inverted CdGM bands, as numerically shown in Fig. 1e. The inverted bands with $J_z = \pm 1$ feature a pair of rotation-protected band crossings, forming a nodal vortex state. The $J_z = 0$ bands, however, will open up a topological gap as the $v_z$ term of $h^{(1)}$ is included [see Fig. 1f], which forms a Majorana-carrying Kitaev vortex. Moreover, this exotic vortex-line physics holds in the isotropic limit as well, which we confirm numerically by mapping out the vortex topological phase diagram in Fig. 2. Therefore, we have managed to prove that a superconducting anisotropic or isotropic LSM will simultaneously carry topological Kitaev and nodal vortices, i.e., $\nu_0 = \mathcal{Q}_1 = 1$, despite the trivial nature of its normal-state electron bands.

As a 4D irrep of the crystalline group, the quadratic band touching of LSM is unstable against lattice strain effects. It is natural to ask about the stability of the LSM-origined vortex topological phases under strain-induced perturbations. Motivated by this, we consider to perturb the original LSM Hamiltonian with two different strain effects described by $\mathcal{H}'_{\text{LSM}} = -\Sigma_{\text{str}}\gamma_5 + \Sigma_{\text{sb}}\gamma_{15}$. In particular, a positive (negative) $\Sigma_{\text{str}}$ describes a uniaxial tensile (compressive) strain that reduces the original $O(3)$ symmetry to an around-$\hat{z}$ continuous rotation symmetry $C_\infty$. Meanwhile, $\Sigma_{\text{sb}}$ further breaks $C_\infty$ down to a twofold rotation $C_2$. Both terms preserve inversion symmetry $\mathcal{P} = \gamma_0$ of the normal-state Hamiltonian. In Fig. 2, we numerically map out the vortex topological phase diagrams (VTPDs) as a function of $\mu, \Sigma_{\text{str}}$, and $\Sigma_{\text{sb}}$. This is achieved by regularizing the vortex-inserted LSM Hamiltonian $(\mathcal{H}_{\text{LSM}} + \mathcal{H}'_{\text{LSM}})$ on a $80 \times 80$ square latttice and calculating its CdGM energy spectrum along $k_z$. As elaborated in Supplementary Note 4, the VTPDs for lattice-regularized models generally agree well with those of the continuum models in a quantitative manner. Whenever the CdGM gap closes at $k_z = 0$, the vortex-line topology will simultaneously change.

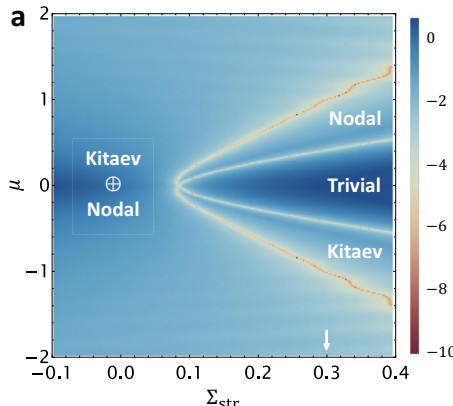
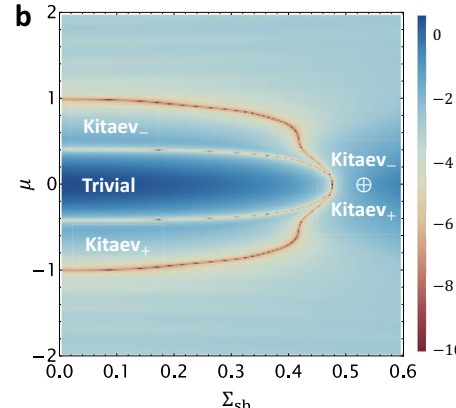

**Fig. 2 | Vortex topological phase diagrams (VTPD) of a strained LSM.** Both VTPDs are mapped out by calculating the vortex-state energy gap at $k_z = 0$, whose logarithmic value is shown by the colors in (**a**) and (**b**). Vortex topology changes whenever the vortex-state gap closes. **a** shows the VTPD as a function of $\Sigma_{str}$ and $\mu$. Specifically, the normal state is a topologically trivial insulator for $\Sigma_{str} > 0$ and a Dirac semimetal for $\Sigma_{str} < 0$. **b** shows the VTPD as a function of $\Sigma_{sb}$ and $\mu$, with a fixed $\Sigma_{str} = 0.3$ [white arrow in (**a**)]. The rotational symmetry breaking induced by $\Sigma_{sb}$ updates the nodal vortex in (**a**) to the Kitaev$_-$ vortex in (**b**). Here ±is used to represent the eigenvalue of the twofold rotational symmetry. The model parameters for both calculations are the same as those in Fig. 1f.

Let us start with the $\Sigma_{str}$-$\mu$ VTPD in Fig. 2a with $\Sigma_{sb} = 0$. At the bulk-band level, $\Sigma_{str} < 0$ creates a new band inversion around $\Gamma$, leading to a Dirac semimetal phase with a pair of linearly dispersing 3D Dirac nodes on the $k_z$ axis[38]. Unlike Na$_3$Bi or Cd$_3$As$_2$, this Dirac semimetal phase does not feature any topological surface state, because of $\mathcal{P} = \gamma_0$. Remarkably, the VTPD is governed by the coexistence of Kitaev and nodal vortex phases (denoted as Kitaev $\oplus$ Nodal) for $\Sigma_{str} \leq 0$, as shown in Fig. 2a. This agrees with our analytical perturbation theory derived in Supplementary Note 3, where a negative $\Sigma_{str}$ enhances the band inversions of CdGM bands and thus stabilizes the Kitaev $\oplus$ Nodal phase. Conversely, a positive $\Sigma_{str}$ would destabilize this phase at small $\mu$. Because $\Sigma_{str} > 0$ energetically shifts the electron bands in the opposite way, driving the system into a trivial band insulator. When $\mu$ lies inside the band gap ($|\mu| < \Sigma_{str}$), the vortex-line topology is guaranteed to be trivial for having neither bulk nor surface states at the Fermi level, further forming a fan-shaped trivial vortex regime as confirmed in Fig. 2a. Strikingly, hole (electron) doping of this trivial insulator will enable a topological Kitaev (nodal) vortex phase.

Switching on $\Sigma_{sb}$ generally spoils symmetry protection of the nodal vortex phase by introducing a topological gap for the CdGM states. Due to the PHS and the remaining $C_2$, this new gapped vortex state necessarily carries a nontrivial Kitaev $\mathbb{Z}_2$ index $\nu_1 = 1$ in the $C_2 = -1$ sector. Therefore, this $\Sigma_{sb}$-induced Kitaev phase is topologically distinct from the preexisting Kitaev vortex phase that carries $\nu_0 = 1$, a manifestation of the $C_2$-stabilized $\mathbb{Z}_2 \times \mathbb{Z}_2$ vortex topological classification shown in Table 1. We thus dub a Kitaev vortex phase living in the $C_2 = \pm1$ sector a Kitaev$_\pm$ vortex phase, to highlight its symmetry-eigenvalue label. For a fixed $\Sigma_{str} = 0.3$ (i.e., the normal state is the trivial insulator phase), we numerically map out the $\Sigma_{sb}$-$\mu$ VTPD, as shown in Fig. 2b. Interestingly, the VTPD contains all four gapped vortex phases dictated by the set of $\mathbb{Z}_2 \times \mathbb{Z}_2$ topological indices ($\nu_0, \nu_1$): trivial phase with (0, 0), Kitaev$_+$ phase with (1, 0), Kitaev$_-$ phase with (0, 1), and Kitaev$_- \oplus$ Kitaev$_+$ phase with (1, 1). In Supplementary Note 2.3, we numerically calculate the surface local density of states for both Kitaev$_\pm$ vortex phases using the recursive Green's function method[39]. The existence of vortex Majorana zero mode for each phase is confirmed by the presence of a zero-bias peak at the vortex core center. This unambiguously demonstrates how a variety of vortex-line topologies, as well as their accompanied Majorana modes, can arise from a doped trivial-band insulator with $s$-wave superconductivity.

## Material realization

The LSM-band physics has been experimentally established in HgTe-class materials, including HgTe[40], $\alpha$-Sn[38,41], pyrochlore iridates such as Pr$_2$Ir$_2$O$_7$[42], half-Heusler alloys such as LaPtBi[43], etc. As shown in Fig. 3b, the typical bulk-band structure of HgTe-class materials is well captured by a six-band Kane model, which consists of a pair of $s$-type $\Gamma_6$ electron bands with $J_z = \pm1/2$ and a quartet of $p$-type $\Gamma_8$ hole bands with $J_z = \pm1/2$ [light holes (LHs)] and $J_z = \pm3/2$ [heavy holes (HHs)]. To achieve LSM bands, the band order between $\Gamma_6$ and LH-bands needs to be inverted when compared to that in semiconductors such as CdTe. This band inversion makes $\Gamma_6$ and LHs a typical TI band set, sitting right below the $\Gamma_8$ band touching (i.e., LSM). As a result, LSM and TI bands always coexist near the Fermi level in HgTe-class materials, as shown in the surface spectrum of HgTe in Fig. 3c.

Given the Dirac surface state in Fig. 3c, a direct application of the Fu-Kane theory would immediately predict the existence of gapped Kitaev vortex topology in the vortex phase diagram. Such a prediction, however, is oversimplified for dropping both the HH band and the relevant LSM physics. In addition to the TI-induced Kitaev vortex, we expect the $\Gamma_8$ quartet itself will contribute to one additional nodal vortex state, as well as another Kitaev vortex state, following the analysis in Fig. 1. As a result, we predict that HgTe-class material will only host a single nodal vortex instead of a Kitaev one, since

$$\underbrace{\text{Kitaev vortex} \times 2}_{\text{TI} \oplus \text{LSM}} \oplus \underbrace{\text{nodal vortex}}_{\text{LSM}} \equiv \underbrace{\text{nodal vortex}}_{\text{HgTe}}. \tag{4}$$

Here, two Kitaev vortices annihilate each other topologically due to their $\mathbb{Z}_2$ topological classification.

To verify Eq. (4), our strategy is to start with a TI-based vortex system with well-defined Fu-Kane physics, and then gradually turn on the LSM physics to explore the evolution of vortex topology. This motivates us to define a generalized six-band Kane model with a new coupling parameter $\kappa$, which serves as an effective measure of the overall coupling strength between HH bands and the remaining TI bands. In particular, we have

$$\mathcal{H}_{\text{Kane}}(\kappa, \mathbf{k}) = \begin{pmatrix} h_{\text{TI}}(\mathbf{k}) & \kappa T(\mathbf{k}) \\ \kappa T^\dagger(\mathbf{k}) & h_{\text{HH}}(\mathbf{k}) \end{pmatrix}. \tag{5}$$

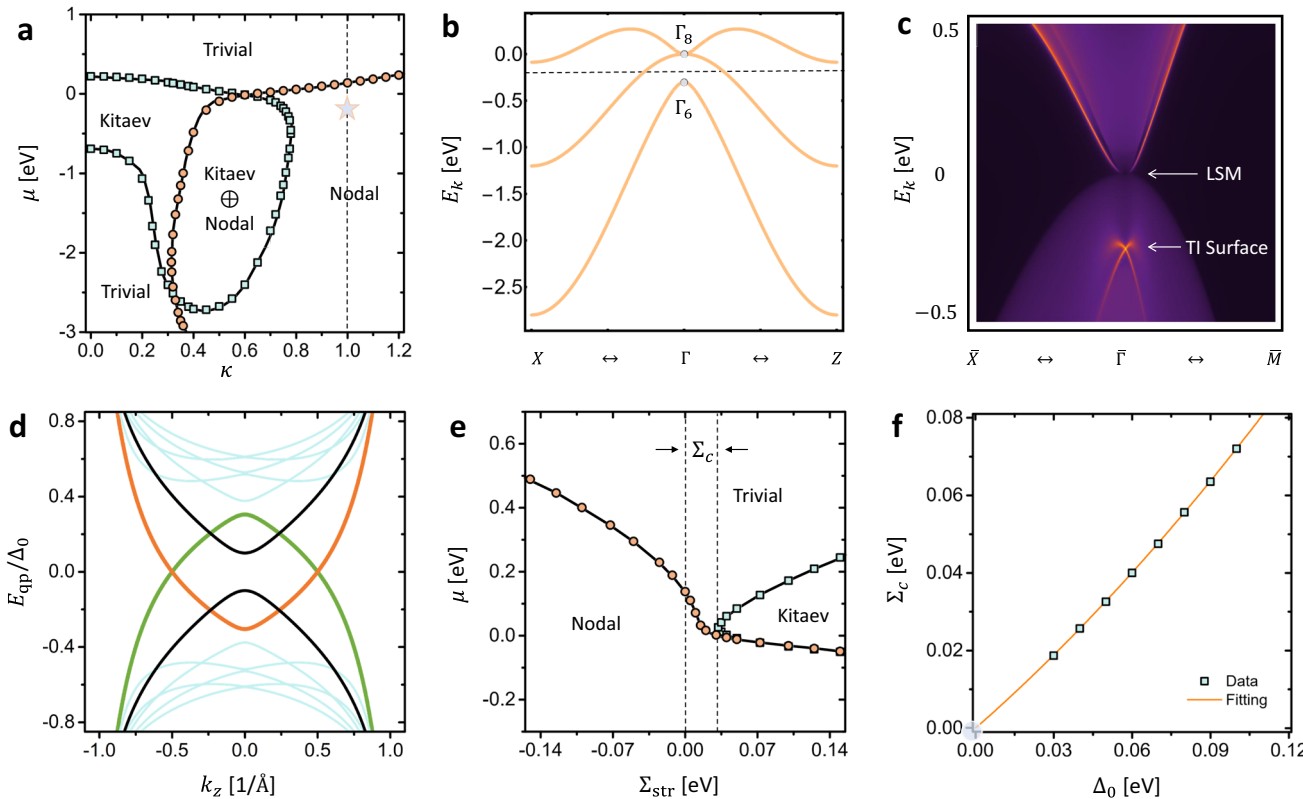

**Fig. 3 | Vortex phase diagram of HgTe.** In (**a**), we show the phase diagram as a function of inter-band coupling $\kappa$ and the chemical potential $\mu$, which includes Kitaev vortex (small $\kappa$), Kitaev $\oplus$ nodal vortex (intermediate $\kappa$), nodal vortex (large $\kappa$) and trivial vortex. $\kappa = 1$ is the Luttinger semimetal (LSM) limit, which recovers the realistic model parameters for HgTe (dashed black line). **b**, **c** Show the bulk and (001) surface dispersions of HgTe based on a realistic six-band Kane model, which clearly reveals the coexisting topological insulator (TI) and LSM physics. In (**d**), the nodal vortex spectrum $E_{qp}/\Delta_0$ is calculated for the star location in (**a**), with two bands carrying $J_z = -1$ (orange line) and $J_z = +1$ (green line) crossing at zero energy. The strain-controlled topological phase diagram is shown in (**e**) as a function of the strain strength $\Sigma_{str}$ and $\mu$, where the critical strain strength $\Sigma_c$ is defined. **f** shows the scaling behavior of $\Sigma_c$ as a function of $\Delta_0$. The fitting function in the orange dashed line is exactly extrapolated to the origin.

The TI bands are described by $\mathcal{H}_{TI} = E_+\gamma_0 + E_-\gamma_{12} + \upsilon/\sqrt{6}(k_y\gamma_{24} - k_x\gamma_{23} + 2k_z\gamma_{25})$. We also denote $h_{HH} = E_8 s_0$ and $E_\pm = (E_6 \pm E_8)/2$, with $E_6 = E_c + \lambda_3 k^2$ and $E_8 = \lambda_1 k^2 - \lambda_2(k_x^2 + k_y^2 - 2k_z^2)$. Controlled by $\kappa$, the inter-band-coupling term is given by

$$\frac{T(\mathbf{k})^\dagger}{\sqrt{3}\lambda_2} = \begin{pmatrix} 0 & -\frac{\upsilon}{\sqrt{6}\lambda_2}k_+ & -k_+^2 & 2k_z k_+ \\ -\frac{\upsilon}{\sqrt{6}\lambda_2}k_- & 0 & -2k_z k_- & -k_-^2 \end{pmatrix}. \quad (6)$$

Notably, the limit with $\kappa = 0$ turns off all the couplings between HH bands and TI bands, which is dubbed a decoupling limit. As $\kappa$ increases, LSM physics is gradually turned on among the $\Gamma_8$ bands until it eventually reaches the isotropic limit of LSM at $\kappa = 1$, which is dubbed the LSM limit. Without loss of generality, we choose the realistic parameter set of bulk HgTe[40] in all our numerical simulations below. Other members in the HgTe-class will have slightly different model parameters, which will only quantitatively, but not qualitatively, modify our phase diagram of the topological vortices.

The VTPD of HgTe with an isotropic s-wave spin-singlet pairing is mapped out as a function of $\kappa$ and the chemical potential $\mu$ in Fig. 3a. The vortex physics of $\mathcal{H}_{Kane}(\kappa, \mathbf{k})$ is numerically simulated in a disk geometry with the Bessel function expansion technique (see Methods). In the decoupling limit $\kappa = 0$, only the Kitaev vortex phase is found in the VTPD for $\mu \in [-0.69 \text{ eV}, 0.22 \text{ eV}]$, which exists around the energy window of the topological gap between $\Gamma_6$ and LH bands. Since the TI physics dominates at $\kappa = 0$, the appearance of a Kitaev vortex

agrees well with both the Fu-Kane theory and the $\pi$-Berry-phase criterion in ref. [10]. As we increase $\kappa$ from zero, the Kitaev vortex region expands rapidly[24] and suddenly vanishes at $\kappa = 0.779$. This observation of Kitaev-vortex cancellation matches our expectation in Eq. (4).

Meanwhile, a new topological region with the nodal vortex starts to emerge at $\kappa = 0.314$ and continues to expand as $\kappa$ grows. Finally, in the isotropic LSM limit with $\kappa = 1$ [i.e., the dashed line in Fig. 3a], only a nodal vortex phase is found in the $\kappa$-$\mu$ VTPD for a large range of $\mu$, in excellent agreement with our prediction in Eq. (4). Nodal vortex dispersion with $\kappa = 1$ and $\mu = -0.15$ eV is shown in Fig. 3d, which clearly illustrates a pair of 1D Dirac points formed by the $J_z = \pm 1$ CdGM states. We further find this nodal vortex state indicated by $\mathcal{Q}_1 = -1$, confirming its topological stability. Note that $\mathcal{Q}_1 = 1$ in Fig. 1 is due to a different parameter choice in the LSM model, which we elaborate in Supplementary Note 2.1. Therefore, despite the fact that HgTe is a zero-gap TI, our calculation predicts a topological nodal phase to show up in its superconducting vortices. This deviation from existing TI-based Majorana vortex paradigms is a direct consequence of trivial-band-induced vortex topology.

**Strain-controlled Majorana engineering**

Given the richness of topological physics in the strain-controlled VTPDs for LSM, we are motivated to explore the physical consequence of perturbing the six-band Kane-model system in Eq. (5) with similar lattice strains. An experimentally relevant in-plane strain effect is described by $\mathcal{H}_{str} = \text{diag}[0, 0, \Sigma_{str}, \Sigma_{str}, -\Sigma_{str}, -\Sigma_{str}]$[38]. This coincides

with the $\Sigma_{str}$ perturbation considered earlier for LSM, and we thus adopt the same notation here.

In Fig. 3e, we numerically map out the VTPD as a function of the strain parameter $\Sigma_{str}$ and $\mu$. The LSM limit $\kappa = 1$ is imposed to match the realistic parameters of HgTe. Similar to the scenario of LSM, a compressive strain with $\Sigma_{str} < 0$ creates a new band inversion between LH and HH bands. This drives the $\Gamma_8$ bands into a 3D Dirac semimetal state with a pair of linear Dirac nodes, coexisting with the $\Gamma_6$-LH TI state[38]. Interestingly, as shown in Fig. 3e, such a compressive strain will lead to a rapid expansion of the nodal vortex region, while no Kitaev vortex phase shows up for any value of $\mu$, similar to the zero-strain limit. Thus, a compressive strain appears to further stabilize the LSM-induced vortex topological physics, instead of spoiling it, which agrees with our LSM-based VTPD in Fig. 2.

A tensile strain with $\Sigma_{str} > 0$ allows LH and HH bands to detach from each other. In this case, the HH bands behave as a set of trivial bands floating inside the topological gap formed by $\Gamma_6$ and LH bands, without touching any of them. Notably, the TI surface state is now the only electron state inside the strain-induced energy gap $E_g \sim 2\Sigma_{str}$ between LHs and HHs. Inside this energy window $E_g$, we expect an emergence of the Kitaev vortex as required by the Fu-Kane paradigm. Indeed, Fig. 3e shows a fan-shaped Kitaev-vortex dome for $\Sigma_{str} > 0$, exactly around $E_g$. Right below the Kitaev-vortex dome, the LSM-induced nodal vortex state remains to be the dominating vortex phase. Together with the $\Sigma_{str}$-$\mu$ VTPD in the compressive region, we conclude that the LSM-induced vortex topological physics is robust against the lattice strain effect, even though the bulk LSM bands are not.

Remarkably, the Kitaev-vortex dome shows up only after a finite positive critical strain $\Sigma_c$ [i.e., the distance between two black dashed lines in Fig. 3e]. While Fu-Kane theory predicts a Kitaev vortex region for an arbitrarily small $\Sigma_{str} > 0$, violation of the Fu-Kane theory occurs when $0 < \Sigma_{str} < \Sigma_c$. We remark that this interesting discrepancy arises from the breakdown of the weak-pairing limit in our numerical simulation, which, however, appears as a basic assumption in the Fu-Kane theory. Specifically, the region where the Fu-Kane picture gets violated in the $\Sigma_{str}$-$\mu$ VTPD is also where both $\Sigma_{str}$ and $\mu$ are smaller than the numerical value of SC order parameter $\Delta_0 = 0.05$ eV in our calculation. Practically, the strong finite-size effect makes it challenging to scale the value of $\Delta_0$ down to a realistic experimental value (e.g., 1 meV) in our simulation. Therefore, it is exactly this finite-pairing effect that allows us to deviate from the Fu-Kane theory. When $\Sigma_{str} > \Delta_0$, we start to approach the weak-pairing limit and this is why the Kitaev-vortex physics begins to show up, signaling a recovery of the Fu-Kane physics.

To eliminate this finite-pairing effect and further test the limit of the Fu-Kane theory, we carry out a careful scaling analysis of $\Sigma_c$ as a function of $\Delta_0$. As shown in Fig. 3f, the scaling relation fits nicely to a simple quadratic relation that is well extrapolated to the origin with $\Sigma_c = \Delta_0 = 0$,

$$\Sigma_c = \chi_1 \Delta_0 + \chi_2 \Delta_0^2, \tag{7}$$

where $\chi_1 = 0.59$ and $\chi_2 = 1.31$ meV$^{-1}$. Physically, the scaling relation implies a monotonic shrink of the Fu-Kane-violation region as the pairing amplitude $\Delta_0$ decreases. When the weak-pairing limit is reached at $\Delta_0 \rightarrow 0^+$, the Fu-Kane limit is fully restored with $\Sigma_c \rightarrow 0^+$. Crucially, we note that $\Delta_0$ is always small but finite in realistic superconducting systems. For example, an experimentally relevant $\Delta_0 \sim 1$ meV will lead to $\Sigma_c \sim 0.6$ meV following Eq. (7). This immediately leads to two important experimental consequences:

(i)  The absence of a Kitaev vortex in an unstrained HgTe generally holds for any small but finite $\Delta_0$;
(ii) Vortex MZMs can be recovered via a strain control, and the critical strain triggers $\Sigma_c \sim 0.6$ meV is experimentally accessible[38].

## Experimental signatures

The $\Sigma_{str}$-$\mu$ VTPD in Fig. 3e sheds light on the detection and manipulation of vortex MZMs. By continuously tuning the strain from a compressive type to a tensile type, the vortex of an electron-doped HgTe (e.g., $\mu \sim 0.1$ eV) will undergo a series of vortex topological phase transitions, from Majorana-free nodal and trivial vortices to a Majorana-carrying Kitaev vortex. Consequently, probing the local density of state (LDOS) at the surface vortex core with a scanning tunneling microscope (STM) will reveal a single transition at $\Sigma_c$, after which a zero-bias peak (ZBP) emerges in the tunneling spectrum, as schematically shown in the bottom panel of Fig. 4a.

While a nodal vortex does not carry MZMs, breaking the around-axis rotation symmetry spoils the vortex nodal structure and further leads to a Kitaev vortex[18]. Such a symmetry-breaking effect can be feasibly generated by tilting the applied magnetic field $\mathbf{B}$, or applying an in-plane lattice strain $\Sigma_{sb}$ following $\mathcal{H}'_{LSM}$ defined for LSM [i.e., replacing $\sqrt{3}\lambda_2 k_x k_y$ with $\sqrt{3}\lambda_2 k_x k_y + \Sigma_{sb}$ in Eq. (6)]. We note that most HgTe-class materials respect either a space group $F\bar{4}3m$ (No. 216) or $Fd3m$ (No. 227), the highest-fold rotation symmetry of which is $C_3$ along (111) direction. Perturbing HgTe-class systems with $\Sigma_{sb}$ will directly break $C_3$ down to $C_1$, which admits a single $\mathbb{Z}_2$ index $\nu_0$. This is crucially different from the fully rotational symmetric LSM considered in the previous sections where $\Sigma_{sb}: C_\infty \mapsto C_2$. Following our notation in Fig. 2, we still denote the nodal-origined Kitaev vortex as Kitaev$_-$ and the preexisting Kitaev vortex as Kitaev$_+$ for convenience. However, one should keep in mind that the Kitaev$_\pm$ vortex phases here are topologically indistinguishable due to the lack of $C_2$ symmetry.

By tuning $\Sigma_{str}$, we expect a Kitaev-trivial-Kitaev transition for a finite $\Sigma_{sb}$. As schematically shown in the top panel of Fig. 4a, an MZM-induced ZBP from the Kitaev$_-$ vortex will first vanish in the LDOS after entering the trivial phase, and will eventually reappear when the Kitaev$_+$ vortex is turned on. This transition for a fixed $\mu = 0.1$ eV is explicitly verified by numerically mapping out the VTPD as a function of $\Sigma_{sb}$ and $\Sigma_{str}$, which we summarize in Fig. 4b. Here, we have regularized the strained HgTe model on a $50 \times 50$ 2D square lattice, while keeping $k_z$ a good quantum number. $\Delta_0 = 0.1$ eV is applied to eliminate any possible finite-size effect. We further numerically explore the VTPD for a fixed $\Sigma_{sb} = 0.2$ eV by varying both $\mu$ and $\Sigma_{str}$ and have observed the same Kitaev-trivial-Kitaev transition, as shown in Fig. 4c.

Finally, we wonder if the Kitaev$_\pm$ phases in HgTe, despite their topological equivalence, could be locally distinguished from each other through surface LDOS measurements. Using the recursive Green's function method, we numerically calculate the spatial spin-resolved surface LDOS $D_\uparrow(\mathbf{r}_\parallel)$ and $D_\downarrow(\mathbf{r}_\parallel)$ at a zero-bias voltage for the strained HgTe model in a semi-infinite geometry along the $\hat{z}$ direction. Open boundary conditions are imposed for both in-plane directions with $N_x = N_y = 35$ and we have chosen $\Sigma_{sb} = 0.2$ eV, $\mu = 0.2$ eV and $\Delta_0 = 0.2$ eV for all calculations to eliminate the in-plane finite-size effect. Here, $\mathbf{r}_\parallel = (x, y)$ and the vortex core center locates at $\mathbf{r}_c = (18, 18)$ in a unit of in-plane lattice constant $a_x = a_y = 6.46$ Å. The spin-resolved LDOS plots for a representative Kitaev$_-$ vortex phase [the white dot in Fig. 4c] are shown in Fig. 4d–f. In particular, $D_\downarrow(\mathbf{r}_\parallel)$ shows a greater ZBP than that of $D_\uparrow(\mathbf{r}_\parallel)$ at $\mathbf{r}_c$. In contrast, the zero-bias spin texture for the Kitaev$_+$ vortex phase [the white square in Fig. 4c] is exactly the opposite, where the ZBP of $D_\uparrow(\mathbf{r}_\parallel)$ is significantly higher than $D_\downarrow(\mathbf{r}_\parallel)$ at $\mathbf{r}_c$. Therefore, a state-of-the-art spin-polarized STM should be capable of extracting the distinct spin patterns for the Kitaev$_\pm$ phases in HgTe-class materials. We further note that the spin pattern for the Kitaev$_-$ phase here is consistent with that of the Kitaev$_-$ vortex phase of LSM [see Fig. 3 of Supplementary Note 2.3], agreeing with the fact that the Kitaev$_-$ phase of the Kane model arises from the overall trivial LSM-dominant bands. Observing the above wavefunction information, together with strain-induced ZBP transitions, will provide rather

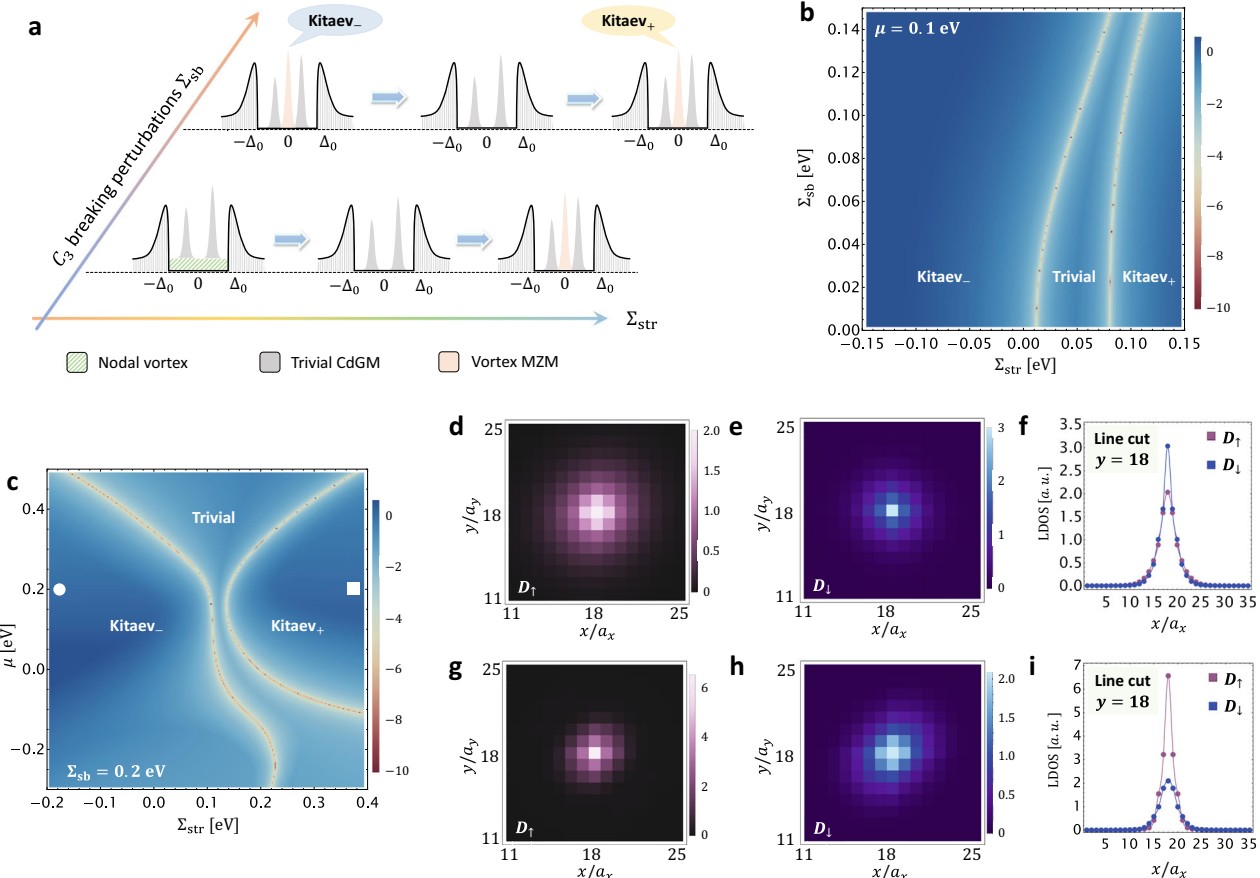

**Fig. 4 | Strain-controlled Majorana engineering of HgTe-class materials.**
**a** Schematically shows the evolution of the local density of state (LDOS) at the vortex core center as a function of bias voltage by tuning both the in-plane lattice strain strength $\Sigma_{str}$ and the $C_3$-symmetry-breaking perturbation $\Sigma_{sb}$. The Kitaev-trivial-Kitaev transition with vortex Majorana zero mode (MZM) of a Kitaev vortex in (**a**) is numerically verified by both mapping the $\Sigma_{str}$-$\Sigma_{sb}$ VTPD in (**b**) at a fixed $\mu = 0.1$ eV, and the $\Sigma_{str}$-$\mu$ VTPD in (**c**) at a fixed $\Sigma_{sb} = 0.2$ eV. The colors in (**b**) and (**c**) represent the logarithmic value of the vortex energy gap at $k_z = 0$. The color map plots of the spatial spin-resolved surface LDOS (a.u.=arbitrary units) at a zero-bias voltage are numerically calculated for the Kitaev$_-$ vortex in (**d**–**f**) and the Kitaev$_+$ vortex in (**g**–**i**), respectively. These two topologically equivalent Kitaev vortex phases can be clearly distinguished by their distinct zero-bias spin textures as shown in **f** with $D_\uparrow < D_\downarrow$ and **i** with $D_\uparrow > D_\downarrow$ at the vortex core center $\mathbf{r}_c = (18, 18)$ in a unit of in-plane lattice constants $a_x$ and $a_y$.

compelling experimental evidence for the Majorana nature of these topological vortices.

## Discussion

We have demonstrated the possibility of topological nontrivial superconducting vortices based on a set of topology-free electronic bands. On the material side, we have established HgTe-class materials as an unprecedented playground to study trivial-band-induced vortex topology. We notice that intrinsic or proximity-induced superconductivity has already been observed in several members of this material family, including HgTe/Nb heterostructure[44], α-Sn/PbTe heterostructure[45,46], and half-Heusler alloys such as LaPtBi[47], YPtBi[48], and RPdBi with R = Lu, Tm, Er, Ho[49]. Our theory will serve as important guidance to detect, control, and engineer Majorana modes in these candidate superconducting systems.

Our results further suggest several new guidelines for the ongoing vortex-based Majorana search. First of all, we note that most topological-band-based SC candidates have coexisting trivial bands near the Fermi level, while most literature chooses to drop the trivial bands to simplify the vortex topology analysis. Our finding, however, suggests that trivial bands in a topological-band SC should have also been in the spotlight, without which the Majorana interpretation of the material could be fallacious. Second, we should not limit the Majorana-oriented material search to intrinsic TSCs or topological-band SCs,

since Majorana vortices can exist in certain types of bulk-topology irrelevant SCs as well. We hope that our work will motivate more theoretical and experimental research efforts under the spirit of Majorana from trivial bands and further initiates a new journey of the Majorana research in this largely uncharted territory, the trivial superconductors.

## Methods

### Bessel function expansion

The Bessel function expansion technique enables the calculation of the vortex energy spectrum for continuum models, which we will describe below. In a rotation-symmetric disk or cylinder geometry, a BdG Hamiltonian $\mathcal{H}_{BdG}$ is characterized by two good quantum numbers, $z$-directional crystal momentum $k_z$ and $z$-component total angular momentum $J_z$. In particular, the angular momentum operator is

$$\hat{J}_z = (-i\partial_\theta)I_{2N_h \times 2N_h} + J_{basis} + J_{vortex}, \quad (8)$$

where $I_{2N_h \times 2N_h}$ is the $2N_h$-by-$2N_h$ identity matrix with $N_h$ the dimension of the normal-state Hamiltonian and $(r, \theta)$ denote the in-plane polar coordinates. For the 4-band LSM ($N_h = 4$), we have

$$J_{basis} = \text{diag}\left[\frac{3}{2}, -\frac{1}{2}, \frac{3}{2}, -\frac{1}{2}, \frac{1}{2}, -\frac{3}{2}, \frac{1}{2}, -\frac{3}{2}\right]. \quad (9)$$

Here, $J_{\text{vortex}}$ arises from the vortex phase winding,

$$J_{\text{vortex}} = \text{diag}\left[-\frac{1}{2}, \frac{1}{2}, \frac{1}{2}, -\frac{1}{2}, -\frac{1}{2}, \frac{1}{2}, \frac{1}{2}, -\frac{1}{2}\right]. \quad (10)$$

Clearly, $[\hat{J}_z, \mathcal{H}_{\text{BdG}}] = 0$, and the BdG Hamiltonian matrix can be decomposed into $J_z$-labeled matrix blocks,

$$\mathcal{H}_{\text{BdG}} = \sum_{J_z} \oplus H_{J_z}(r,\theta). \quad (11)$$

As a result, we only need to solve $H_{J_z}(r,\theta)|\Phi(J_z,r,\theta)\rangle = E|\Phi(J_z,r,\theta)\rangle$, where a general energy eigenstate is $J_z$ labeled and further takes the following form,

$$\begin{aligned}|\Phi(J_z,r,\theta)\rangle = e^{i(J_z-1)\theta}[&u_1(J_z-1,r), u_2(J_z,r)e^{i\theta},\\ &u_3(J_z-2,r)e^{-i\theta}, u_4(J_z+1,r)e^{2i\theta}, v_1(J_z,r)e^{i\theta},\\ &v_2(J_z+1,r)e^{2i\theta}, v_3(J_z-1,r), v_4(J_z+2,r)e^{3i\theta}]^T,\end{aligned} \quad (12)$$

where both $u_i(n, r)$ and $v_i(n, r)$ with $i = 1, 2, 3, 4$ yield the following expansions,

$$u(J_z,r) = \sum_{j=1}^{N} c_{jJ_z}\phi(J_z,r,\alpha_j), \quad (13a)$$

$$v(J_z,r) = \sum_{j=1}^{N} c'_{jJ_z}\phi(J_z,r,\alpha_j). \quad (13b)$$

Here, the normalized Bessel function is defined as

$$\phi(J_z,r,\alpha_i) = \frac{\sqrt{2}}{R}\mathcal{J}_{J_z}(\alpha_i r/R)/\mathcal{J}_{J_z+1}(\alpha_i), \quad (14)$$

where $\mathcal{J}_n$ is the Bessel function of the first kind. $\alpha_i$ and $R$ denote the $i^{\text{th}}$ zero of $\mathcal{J}_{J_z}(r)$ and the radius of the disk, respectively. $c$ and $c'$ are expansion coefficients that are yet to be numerically calculated. We further note that in the polar coordinate system, the crystal momenta $k_{\pm} = k_x \pm ik_y$ become

$$k_+ = e^{i\theta}\left[-i\frac{\partial}{\partial r} + \frac{1}{r}\frac{\partial}{\partial \theta}\right], \quad (15a)$$

$$k_- = e^{-i\theta}\left[-i\frac{\partial}{\partial r} - \frac{1}{r}\frac{\partial}{\partial \theta}\right], \quad (15b)$$

which satisfy

$$k_+(e^{in\theta}\mathcal{J}_n(\alpha r)) = i\alpha e^{i(n+1)\theta}\mathcal{J}_{n+1}(\alpha r), \quad (16a)$$

$$k_-(e^{in\theta}\mathcal{J}_n(\alpha r)) = -i\alpha e^{i(n-1)\theta}\mathcal{J}_{n-1}(\alpha r). \quad (16b)$$

It is also easy to show that

$$\left(k_x^2 + k_y^2\right)\left[e^{in\theta}\mathcal{J}_n(\alpha r)\right] = \alpha^2\left[e^{in\theta}\mathcal{J}_n(\alpha r)\right]. \quad (17)$$

The energy eigen-equation is now essentially a set of 1D radial equations for fixed $k_z$ and $J_z$. In addition, the disk geometry with hard-wall boundary conditions requires $|\Phi(J_z,r,\theta)\rangle$ to satisfy $u_i(r=R) = v_i(r=R) = 0$. Notably, a Bessel functions with a large $\alpha_i$ will oscillate rapidly and we expect it to contribute little to the low-energy vortex-bound states. Therefore, for a reasonably large $N \in \mathbb{Z}_{>0}$, we can truncate the zeros of the Bessel functions at $\alpha_N$, making the dimension of

each decoupled Hilbert subspace to be $8N$. Physically, this truncation can be interpreted as a Debye frequency cutoff around the Fermi energy. Solving these radial equations leads us to the vortex-bound states and their energy relations for a general vortex problem.

The vortex simulation of the LSM model in the continuum limit is performed using the above Bessel function expansion technique with $R_{\text{disk}} = 250$. We further truncate the zeros of the Bessel function at $N = 250$ and numerically confirm the validity of this truncation. As discussed in Supplementary Note 4, the continuum model approach agrees quantitatively with the discrete tight-binding model approach.

As for the 6-band Kane model ($N_h = 6$), a general vortex wavefunction that respects the rotation symmetry is given by

$$\begin{aligned}|\Phi_{\text{Kane}}(J_z,r,\theta)\rangle = e^{iJ_z\theta}[&u_1(J_z,r), u_2(J_z+1,r)e^{i\theta},\\ &u_3(J_z,r), u_4(J_z+1,r)e^{i\theta}, u_5(J_z+2,r)e^{2i\theta},\\ &u_6(J_z-1,r)e^{-i\theta}, v_1(J_z,r), v_2(J_z-1,r)e^{-i\theta},\\ &v_3(J_z,r), v_4(J_z-1,r)e^{-i\theta}, v_5(J_z-2,r)e^{-2i\theta},\\ &v_6(J_z+1,r)e^{i\theta}]^T,\end{aligned} \quad (18)$$

where the components $u_i(J_z, r)$ and $v_i(J_z, r)$ with $i = 1, 2, \ldots, 6$ can be both expanded by the normalized Bessel functions, as we discussed earlier. To eliminate the finite-size effect that is induced by a small $\Delta_0$, we consider a large disk radius of $R_{\text{disk}} = 2100$ in units of the in-plane lattice constant. The truncation of the zeros of the Bessel function is $N = 385$ and the dimension of Hilbert space in our simulation is $12N = 4620$.

We finally remark on the particle-hole symmetry $\Xi$ of $|\Phi(J_z,r,\theta)\rangle$. Starting from an eigenstate at $k_z = 0$ with $H_{J_z}|\Phi(J_z,r,\theta)\rangle = E_{J_z}|\Phi(J_z,r,\theta)\rangle$, we have

$$|\Phi'(-J_z,r,\theta)\rangle = \Xi|\Phi(J_z,r,\theta)\rangle, \quad (19a)$$

$$H_{J_z}|\Phi'(-J_z,r,\theta)\rangle = -E_{J_z}|\Phi'(-J_z,r,\theta)\rangle. \quad (19b)$$

Since our continuum models with isotropic $s$-wave spin-singlet pairings feature a full rotation symmetry, the $J_z = 0$ subspace $H_{J_z=0}$ is the only sector that respects particle-hole symmetry, while a $J_z \neq 0$ subspace is related to the $-J_z$ one via particle-hole symmetry.

## Chiral winding number and vortex zero modes

We discuss the winding number argument to understand the existence of vortex zero modes of LSM in Fig. 1c. As shown in Supplementary Note 2.2, it is suggestive to separate Eq. (2) into a direct sum of two matrix blocks $H_0 = h_\Delta(\mathbf{k}_\parallel, \theta) \oplus h_{-\Delta}(\mathbf{k}_\parallel, \theta)$ and a perturbation part $H_1(\mathbf{k}_\parallel, k_z)$. In particular,

$$\begin{aligned}h_\Delta(\mathbf{k}_\parallel, \theta) = \quad &\Delta(\cos\theta\tau_x\sigma_0 - \sin\theta\tau_y\sigma_z)\\ &+ \tilde{v}\left[-\left(k_x^2 - k_y^2\right)\tau_y\sigma_y + 2k_xk_y\tau_y\sigma_x\right].\end{aligned} \quad (20)$$

It is easy to check that $h_\Delta(\mathbf{k}_\parallel, \theta)$ respects an emergent chiral symmetry

$$\mathcal{S} = \tau_z\sigma_0, \quad (21)$$

which is independent of the sign of $\Delta$. A stable vortex zero mode is necessarily an eigenstate of $\mathcal{S}$ and carries a $\mathcal{S} = \pm 1$ label. Only zero modes that are differently $\mathcal{S}$-labeled can interact with each other and get hybridized, while those carrying the same label cannot get coupled.

Now $h_\Delta(\mathbf{k}_\parallel, \theta)$ manifests as an effective 3D Hamiltonian in the symmetry class AIII, whose topological behavior is characterized by a

chiral winding number $\mathcal{N}_S \in \mathbb{Z}$[36]. Physically, we have

$$\mathcal{N}_S = \mathcal{N}_{+1} - \mathcal{N}_{-1}. \tag{22}$$

Here $\mathcal{N}_{\pm 1}$ denotes the number of vortex zero modes that carry $S = \pm 1$. Evaluation of $\mathcal{N}_S$ can be achieved by noting that $h_\Delta(\mathbf{k}_\parallel, \theta)$ yields an off-block-diagonal form, as a result of the chiral symmetry,

$$h_\Delta(\mathbf{k}_\parallel, \theta) = \begin{pmatrix} 0 & Q(\mathbf{k}_\parallel, \theta) \\ Q^\dagger(\mathbf{k}_\parallel, \theta) & 0 \end{pmatrix}, \tag{23}$$

with

$$Q(\mathbf{k}_\parallel, \theta) = \begin{pmatrix} \Delta e^{i\theta} & \bar{v} k_-^2 \\ -\bar{v} k_+^2 & \Delta e^{-i\theta} \end{pmatrix}. \tag{24}$$

Then the chiral winding number can be written as

$$\mathcal{N}_S = -\frac{1}{24\pi^2} \int d^2\mathbf{k}\, d\theta\, \epsilon^{\mu\nu\rho} \operatorname{Tr}\left[(Q\partial_\mu Q^\dagger)(Q\partial_\nu Q^\dagger)(Q\partial_\rho Q^\dagger)\right], \tag{25}$$

where $\mu, \nu, \rho \in \{k_x, k_y, \theta\}$ and $\epsilon^{\mu\nu\rho}$ is the Levi-Civita tensor. Applying Eq. (25) to Eq. (24), we arrive at

$$\begin{aligned} \mathcal{N}_S &= -\frac{1}{24\pi^2} \int_0^{2\pi} d\varphi \int_0^{2\pi} d\theta \int_0^\infty \frac{48 \bar{v}^2 \Delta^2 k^2}{(\bar{v}^2 k^4 - \Delta^2)^2} k\, dk \\ &= -\frac{1}{24\pi^2} (2\pi)^2 (-12) = 2. \end{aligned} \tag{26}$$

Similarly, $\mathcal{N}_S = 2$ also holds for the other $4 \times 4$ block $h_{-\Delta}$ since the value of $\mathcal{N}_S$ is independent of the sign of $\Delta$. As a result, the net chiral winding number for $H_0$ is

$$\mathcal{N}_S^{(\text{net})} = 4, \tag{27}$$

indicating four robust zero-energy vortex-bound states with $S = +1$. Projecting $H_1(\mathbf{k}_\parallel, k_z)$ onto the zero-mode basis will lead us to a perturbative understanding of the nontrivial vortex topology in superconducting LSM systems, as illustrated in Fig. 1. The zero modes further serve as the basis for building an analytical perturbation theory for the vortex-line Hamiltonian of LSM, as shown in the Supplementary Note 3.

## Data availability
The datasets generated during this study are available from the corresponding author on reasonable request.

## Code availability
The custom codes generated during this study are available from the corresponding author on reasonable request.

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

## Acknowledgements
We are grateful to L.-Y. Kong, X.-Q. Sun, J. Yu, J.-S. Lee, B. Seradjeh, P. Ghaemi, T. Hughes, and C. Batista for stimulating discussions. We are particularly indebted to J.-D. Sau for his valuable comments and insight that motivate us to study the scaling relation of the superconducting order parameter. This work is supported by a start-up fund at the University of Tennessee.

## Author contributions
Both authors contributed essentially to the formulation and theoretical analysis of the problem and to writing the manuscript. L.-H.H. performed numerical calculations with the help of R.-X.Z.

## Competing interests
The authors declare no competing interests.
