## [Peer Review File · Nature Communications]

Topological Superconducting Vortex From Trivial Electronic BandsREVIEWER COMMENTS

Reviewer #1 (Remarks to the Author):

In this manuscripts, Hu & Zhang performed theoretical studies on the vortex states behavior of a topological trivial Luttinger semimetal (LSM). Benefit from this insightful studies, they demonstrate convincingly that the emergent nontrivial topology of vortex line is not related to the bulk-state topology, on the other words, vortex Majorana zero modes (MZM) require neither topological band structure nor nontrivial superconducting order parameters. Armed with a generic symmetry discussion on vortex line topology, the authors show that the crystalline symmetries force vortex states into different sectors labeled by angular momentum. A MZM-carrying Class-D case can be defined in the sectors with PHS invariant, while a nodal Class-A case in others. This symmetry-protected topology indicates the absence of one-to-one correspondence between bulk states and vortex. Enlighted by this, the authors discovered that the topological trivial LSM model support a mixing phase of Kitaev and nodal vortex, and a typical material, i.e. HgTe, was studied carefully. The authors verified a robust topological phase in their phase diagram, in which the MZM could be engineered by tuning the LSM bands or general symmetry. Thus this manuscripts delivered a comprehensive study on the problem of vortex topology in a general superconductor, covering from the aspect of fundamental theory, to material realization. It not only provides a deeper understanding on the vortex states, but also a useful guidance for experimental research on this ongoing topic. I am delighted to recommend to publish this manuscripts on Nature Communications after the authors respond to the several minor questions as listed below.

1. It looks like the C_n -Symmetry argument is only for 3D case. Can the authors comment on the effects of crystalline symmetry (C_1 to C_6) on the 2D case? Is there any correspondence between the C_n index J_z to the angular momentum index of integer or half-integer sequences of vortex bound states in 2D?
2. For the conclusion of the LSM vortex, the emergent chiral symmetry is very important to make sure four-fold degeneracy of MZM in $H_0(k//)$. It is better to explain carefully how this chiral symmetry emergent, and what is special in the LSM model. What will happen if the 4-fold MZM hybridize with each other? Does the coexistence of Kitaev and nodal vortex line still hold?
3. Since the authors write "the vortex line topology here manifests a distinct origin from known vortex Majorana theories", it is better to explain the difference between LSM vortex shown here and the trivial DSM vortex studied in Ref. 13.
4. I understand this may beyond the scope of the manuscripts, can author comment on the origin of the anomalous spin behavior of type-A MZM shown in Fig. 3b-c. This is quite different from both nanowire case and Fu-Kane vortex.
5. It is better to cite two more papers, they also focus on the emergent vortex topological invariant, but with different ideas. PRL 118, 147003 (2017); PRL 119, 047001 (2017).
6. A typo: The second to the last line of Fig.3 caption, (c) should be (e).

Reviewer #2 (Remarks to the Author):

The paper presents a theoretical investigation of the band structure of vortex bound states in superconductors. It is demonstrated that the vortex bound states may be topologically non-trivial despite the topologically trivial nature of the bulk state. The results are illustrated by model calculations for HgTe-class materials using experimentally motivated parameters. It is proposed that lattice strain may be utilized to create or destroy Majorana zero modes in these materials.

Majorana quasiparticles in superconductors are widely investigated in the literature. The experimental observations of proposed Majorana zero modes are often accompanied by overly simplified theoretical models which only consider the topology of a single electronic band in most cases, leading to heated

discussions concerning possible alternative explanations. The present manuscript makes an important step by highlighting the role of topologically trivial bands in the formation of topologically non-trivial vortex bound states, indicating that these bands cannot be neglected in a correct theoretical description. The symmetry considerations are supported by detailed numerical calculations using parameters for an experimentally studied class of materials, proposing a new research direction to observe Majorana zero modes in systems with a trival band topology. Therefore, I am convinced that the manuscript will attract the interest of both experimental and theoretical groups working in the field, and recommend it for publication in Nature Communications after the authors clarified a few remaining open questions.

My questions and comments are the following:

1. In the manuscript I could not find a conclusive answer to the question whether Majorana bound states without nodal vortex states may be formed only from topologically trivial bulk bands. In Secs. II and III they appear simultaneously, meaning that the Majorana state at zero energy appears on top of the continuum of the nodal vortex band, and they could hybridize with each other if the rotational symmetry were broken. In Secs. IV and V, it appears that the main effect of the topologically trivial bands is to destroy the Majorana state created by the topologically non-trivial bands. The Majorana state may be restored by strain or symmetry breaking, but it still seems to require the presence of the topologically non-trivial bands in the six-band Kane model.

Although the manuscript is very carefully written and I could not find a statement that would indicate the presence of Majorana states without nodal vortex states in the topologically trivial regime of the bulk, a cursory look over the paper may arrive at the conclusion that Majorana states would be equally simple to realize in topologically trivial as in topologically non-trivial bulk bands. Therefore, I think it would be necessary to clarify this question, for example by explaining whether any of the Majorana states sketched in Fig. 3(a) stem from an overall topologically trivial bulk band structure.

2. Generally, the eigenstates should be classified according to the irreducible representations of the group. For the point groups C_n these may indeed be described by the angular momentum quantum numbers $J_z=0, \dots, n-1$, as shown in Table I. However, including additional mirror symmetries, for example in the C_{nv} groups, will combine some angular momentum quantum numbers into a multi-dimensional irreducible representation, inducing degeneracies in the high-symmetry points. Would these additional symmetries influence the conclusions about the emergence of Kitaev and nodal vortex states? This could be relevant since the point groups O_h and T_d mentioned for the investigated material class in the first paragraph of Sec. III also contain mirrorings.

The model considered in Sec. III has a continuous rotational symmetry around the z axis, which is broken in real systems by the crystal structure. Do the conclusions change if such symmetry-breaking terms are taken into account?

In summary, I think it would be beneficial to discuss in more detail which symmetries are necessary and which symmetries are not allowed for the conclusions to hold.

3. The breaking of the rotational symmetry in Sec. VI, Fig. 3 and Supplementary Fig. 3 is rather vaguely explained. It would be useful to add what was exactly the symmetry-breaking term and which parameter values were used in the calculations illustrated by the spin-resolved LDOS of the zero-energy states.

I think it would also be useful to shortly clarify what the spin-up and spin-down components in the wave function mean, since the bulk Hamiltonian is invariant under time reversal where spin-up and spin-down states should be equivalent. Are the spin-up and spin-down components connected to the sign of the phase winding in the pairing function around the vortex?

Other minor comments:

4. In the first paragraph on page 4, "positive $m_{\{1\}}$ " and "negative $m_{\{2\}}$ " are mentioned, which contradicts the signs shown in Fig. 1.

5. In the Hamiltonian defined in Eqs. (20) and (21) of the Supplementary Material, the pairing potential seems to connect spin-up electrons with spin-up holes, although it describes s-wave pairing. Does the spin-up hole denote the annihilation of a spin-down electron? In the text, it is only highlighted that the e and h indices describe opposite wave vectors.

6. In Eq. (21) in the Supplementary Material, the connection of the v and \tilde{v} parameters to the parameters in the Hamiltonian in Eq. (16) seems not be explained.

7. In the line above Eq. (31) on page 6 of the Supplementary Material, $k_{\{x\}}+k_{\{y\}}$ should probably be replaced by $k_{\{x\}}+ik_{\{y\}}$.

8. In the first paragraph of Supplementary Sec. IID on page 3, the $m_{\{1\}}$, $m_{\{2\}}$ and $v_{\{z\}}$ parameters used for the simulation appear to be missing.

Reviewer #1

In this manuscripts, Hu & Zhang performed theoretical studies on the vortex states behavior of a topological trivial Luttinger semimetal (LSM). Benefit from this insightful studies, they demonstrate convincingly that the emergent nontrivial topology of vortex line is not related to the bulk-state topology, on the other words, vortex Majorana zero modes (MZM) require neither topological band structure nor nontrivial superconducting order parameters. Armed with a generic symmetry discussion on vortex line topology, the authors show that the crystalline symmetries force vortex states into different sectors labeled by angular momentum. A MZM-carrying Class-D case can be defined in the sectors with PHS invariant, while a nodal Class-A case in others. This symmetry-protected topology indicates the absence of one-to-one correspondence between bulk states and vortex. Enlighted by this, the authors discovered that the topological trivial LSM model support a mixing phase of Kitaev and nodal vortex, and a typical material, i.e. HgTe, was studied carefully. The authors verified a robust topological phase in their phase diagram, in which the MZM could be engineered by tuning the LSM bands or general symmetry. Thus this manuscripts delivered a comprehensive study on the problem of vortex topology in a general superconductor, covering from the aspect of fundamental theory, to material realization. It not only provides a deeper understanding on the vortex states, but also a useful guidance for experimental research on this ongoing topic. I am delighted to recommend to publish this manuscripts on Nature Communications after the authors respond to the several minor questions as listed below.

Reply: We sincerely appreciate the reviewer for his/her recommendation of our manuscript.

The reviewer has raised important and inspiring questions that motivate us to revise and improve our manuscript significantly. A summary of the major revisions in the updated manuscript is listed below.

1.) Topological vortex phases for Luttinger semimetal.

- In the revised main text, we add a new figure [see Fig. 2] to explore topological vortex phase diagrams in strained Luttinger semimetal. While a tensile strain drives Luttinger semimetal into a trivial band insulator, quite remarkably, electron and hole doping of this trivial insulator will lead to nodal and Kitaev vortex phases, respectively.
- In the revised supplementary material, we add two new figures [see Fig. 3 & 4] to characterize the Kitaev vortex phase by numerically calculating the total and spin-resolved surface local

density of states (LDOS) by using the iterative Green's function technique. In this 3D geometry calculation, we successfully identified LDOS signals of vortex Majorana zero modes that are generated by doping a trivial band insulator.

2.) Topological vortex phases for the six-band Kane model.

- In the revised main text, we significantly update Fig. 4 (previously as Fig. 3) by including two topological vortex phase diagrams in Fig. 4 (b) and (c), both of which numerically prove the schematic phase diagram in Fig. 4 (a). We have also updated the spin-resolved surface LDOS for vortex Majorana zero modes two types of Kitaev vortices using the recursive Green's function method, as shown in Fig. 4 (d) and (e).

3.) Lattice model v.s. Continuum model

- In the supplementary materials, we have added a new section and a new figure [see Fig. 5 in Supplementary Note 5] to compare the results obtained from lattice and continuum models. Both modeling approaches agree with each other quantitatively.

Besides, at the end of this reply letter, we have appended detailed summaries of changes for both the main text and the supplementary information. With these revisions and updates, we have comprehensively addressed all the comments and questions from both reviewers. We believe that our work is ready for immediate publication in Nature Communications.

Q1: It looks like the C_n -Symmetry argument is only for 3D case. Can the authors comment on the effects of crystalline symmetry (C_1 to C_6) on the 2D case? Is there any correspondence between the C_n index J_z to the angular momentum index of integer or half-integer sequences of vortex bound states in 2D?

Reply: We thank the reviewer for this interesting question!

Yes, the topological classification table [i.e., Table (I) in the main text] aims at classifying 1D vortex-line topology for s-wave spin-singlet 3D superconductors. The specialty of 3D is that even though the bulk superconductor itself is usually topologically trivial with s-wave pairing, the embedded vortex line itself can still be 1D topological.

2D superconductors, however, do not generally feature the above subsystem topological phenomena due to their reduced dimensionalities. As far as we are aware, **vortex Majorana**

modes in a 2D superconductor always indicate that the 2D system is topological superconducting, and a famous example is the Read-Green model. Interestingly, the relationship between C_n symmetry and vortex Majorana modes in general 2D superconductors has been explored by one of the authors in a very recent work ([arXiv: 2208.01652](https://arxiv.org/abs/2208.01652)). It is found that if the superconductor features C_n -protected higher-order topology, its vortex must feature a pair of C_n -stabilized Majorana modes as a response to the bulk higher-order topology. Again, this “bulk-vortex correspondence” relation is a natural consequence of bulk-state topology, which is **fundamentally distinct** from the emergent 1D vortex topology of 3D superconductors, as described in Table I.

For the second question, J_z is the net z-component angular momentum carried by the CdGM vortex bound states, which include the contributions from spin, orbital, vortex phase winding, etc. Therefore, J_z is exactly the same quantum number for labeling vortex bound states in 2D superconductors (e.g., in the Read-Green model). Without any vortex, J_z is half-integer-valued because of the spinful nature of electrons (we only considered spin-orbit coupled superconductors in this work). The vortex insertion effectively shifts J_z from half-integer to integer-valued, and this is why we have only integer-valued J_z sectors in Table I.

For 2D chiral superconductors, it is the same half-integer to integer transition of J_z that makes the vortex carry a single Majorana zero mode. An elaboration on this vortex Majorana picture can be found in a nice review article (<https://doi.org/10.1088/0034-4885/75/7/076501>).

Q2: For the conclusion of the LSM vortex, the emergent chiral symmetry is very important to make sure four-fold degeneracy of MZM in $H_0(k_{||})$. It is better to explain carefully how this chiral symmetry emergent, and what is special in the LSM model. What will happen if the 4-fold MZM hybridize with each other? Does the coexistence of Kitaev and nodal vortex line still hold?

Reply: We thank the reviewer for these interesting questions and suggestions.

The “emergence” of chiral symmetry lies in the fact that it only emerges in $H_0(k_{||}, k_z = 0)$ when we set $\mu = m_1 = m_2 = 0$. This is a fine-tuned point in the parameter space of the LSM model. Practically speaking, if one can find a matrix A that anticommutes with the Hamiltonian matrix, then A is an effective chiral symmetry. For the LSM model that can be written in terms of gamma matrices, identifying such a chiral symmetry is actually quite feasible with the help of Clifford algebra [see details in Supplementary Note 3.1]. In fact, similar chiral symmetry can also possibly emerge in other Dirac-fermion-based systems. For example, a similar emergent chiral symmetry

has assisted the authors of Ref. [12] to identify the analytical wavefunction of a nodal vortex in a superconducting Dirac semimetal.

We hope to highlight that these emergent MZMs are not the surface Majorana modes of a topological Kitaev vortex. Instead, **they are “bulk” MZMs that will disperse along k_z** , which provides a basis to perturbatively construct a low-energy model for quasi-1D topological vortex phases. Therefore, **the fact that these “bulk” MZMs can hybridize with each other is the key to generating the Kitaev and nodal vortices**. Here, let us follow Fig. 1 of the main text to describe this perturbative picture of inducing vortex-line topology:

First, the emergent chiral symmetry appears only when $k_z = 0$ and $\mu = m_1 = m_2 = 0$ (i.e., all the diagonal terms including $M(k_{\parallel}, k_z) = m_1(k_x^2 + k_y^2) + m_2 k_z^2$ and chemical potential μ vanish), as we discussed in Supplementary Note 3.1. And our numerical results in Fig. 1 (c) in the main text clearly confirm the chiral winding number analysis, namely, there exists four emergent bulk Majorana zero modes.

Second, turning on m_1 and m_2 terms will break the chiral symmetry and further make the MZMs disperse along k_z . Based on our numerical results in Fig. 1 (d) and (e), we find a necessary condition (i.e., $m_1 < 0$ and $m_2 > 0$) leads to the inverted vortex-line bands. Vortex bands in the $J_z = 0$ subspace leads to a Kitaev vortex, while the other sector in the $J_z = \pm 1$ subspace generates a nodal vortex. The topological condition $m_1 m_2 < 0$ for topological vortex states have been derived in Supplementary Note 3.2.

This is how the formation of these MZMs and the subsequent spoilage of them eventually lead us to the 1D vortex topology of LSM.

Q3: Since the authors write “the vortex line topology here manifests a distinct origin from known vortex Majorana theories”, it is better to explain the difference between LSM vortex shown here and the trivial DSM vortex studied in Ref. 13.

Reply: We thank the reviewer for this important question!

The trivial DSM in Ref. 13 shares the same basis functions with a LSM model, so that the two Hamiltonians also share lots of similarities. In fact, the trivial DSM phase can be achieved in a compressively strained LSM with $\Sigma_{str} < 0$, as we described in the main text. In the following Fig. R1 (a), we find that the DSM phase with $\Sigma_{str} < 0$ indeed features both nodal and Kitaev vortex phases, which is consistent with the results in Ref. 13. However, in Ref. 13 and many other

relevant works, there exists a common belief that “**having electronic band inversion is quite essential for inducing vortex topology**”. This is why Ref. 13 only explores the band-inverted DSM phase, but fails to reach out to other regions in the phase diagram. One more difference is that we discover a sufficient topological condition $m_1 m_2 < 0$ for topological vortices in the no-band-inversion phase in our work, while Ref. 13 focuses on $m_1 m_2 > 0$.

A key result of our work is that **vortex topology can even exist in a doped trivial insulator with no band inversion. This is how our work is distinguished from earlier paradigms, including Ref. 13.** To be specific, an example of such an insulator is the tensile strained LSM with $\Sigma_{str} > 0$. Remarkably, as shown in Fig. R1 (b), electron and hole doping of this trivial insulator will lead to topological nodal and Kitaev vortex phases, respectively. We have also calculated the surface local density of state (LDOS) for the Kitaev vortex phases, which unambiguously confirms that vortex Majorana zero modes can arise from a doped, superconducting trivial insulator. These new vortex phase diagrams in Fig. R1 and the surface LDOS plots are presented in the main text and in the supplementary Note 2.5, respectively.

Fig. R1 | Vortex topological phase diagrams (VTPD) of a strained LSM. (a) shows the VTPD as a function of Σ_{str} and μ . (b) shows the VTPD as a function of Σ_{sb} and μ , with a fixed $\Sigma_{str} = 0.3$ [white arrow in (a)]. The rotational symmetry breaking induced by Σ_{sb} updates the nodal vortex in (a) to the Kitaev₋ vortex in (b). Here \pm is used to represent the eigenvalue of the two-fold rotational symmetry. The figure is presented in Fig. 2 in the main text.

Q4: I understand this may be beyond the scope of the manuscripts, can author comment on the origin of the anomalous spin behavior of type-A MZM shown in Fig. 3b-c. This is quite different from both nanowire case and Fu-Kane vortex.

Reply: We thank the reviewer for this interesting question!

From the perspective of Bessel function expansion, a MZM wavefunction is a linear superposition of a complete set of Bessel functions. Some of the Bessel functions such as $J_0(r)$ peaks at $r = 0$, while all the others such as $J_1(r)$ peaks at a finite r . Therefore, **whether the spin-resolved LDOS shows a ring-like or a dot-like pattern is completely determined by the weights of these Bessel functions within the Majorana wavefunction**. Unfortunately, there is unlikely to be any symmetry or topological principle that can determine these spin textures for MZMs. As a result, we expect **the spin information for MZMs to be model dependent in general**.

Here, we would like to clarify that our earlier results in Fig. 3b-c are obtained through a simplified perturbative approach. After realizing the limitations of our previous method, we have developed a recursive Green's function technique to accurately calculate the surface LDOS around the surface vortex core. Consequently, we have updated the relevant spin-resolved LDOS plots in the new Fig. 4 (which replaces the previous Fig. 3).

Fig. R2 | Strain-controlled Majorana engineering of HgTe-class materials. (c) shows the Kitaev-trivial-Kitaev transition is numerically verified by mapping the Σ_{str} - Σ_{sb} VTPD at a fixed $\Sigma_{sb} = 0.2$ eV. The spatial spin-resolved surface LDOS at a zero-bias voltage is numerically calculated for the Kitaev₊ vortex in (d) - (f) and the Kitaev₋ vortex in (g) - (i), respectively. These two topologically equivalent Kitaev vortex phases can be clearly distinguished by their distinct zero-bias spin textures as shown in (f) with $D_{\uparrow} < D_{\downarrow}$ and (i) with $D_{\uparrow} > D_{\downarrow}$ at the vortex core center $\vec{r}_c = (18,18)$. This figure is presented in Fig. 4 in the main text.

The revised spin resolved LDOS D_{\uparrow} and D_{\downarrow} at a zero-bias voltage for both Kitaev_± vortex states of the Kane model with HgTe parameters are numerically calculated and shown in the above Fig. R2. We note that the ring-like pattern in our earlier results is absent in the updated calculation. Nonetheless, the two Kitaev_± phases still display distinct spin textures, where $D_{\uparrow}(r = 0) > D_{\downarrow}(r = 0)$ for the Kitaev₋ MZM and $D_{\uparrow}(r = 0) < D_{\downarrow}(r = 0)$ for the Kitaev₊ MZM. In experiments,

this distinct spin pattern should still be visible with the help of state-of-the-art spin-polarized scanning tunneling microscopy.

We note that it is still possible that the ring-like pattern indeed exists in the LDOS calculations of Fig. R2, but they are simply invisible due to the spatial resolution of our simulation. For example, if the radius of the ring is smaller than the lattice constant of the model, our Green's function approach is incapable of visualizing such a "sub-lattice" spatial feature.

Finally, let's briefly comment on the spin textures for both the Fu-Kane vortex and nanowires:

- (i) For a Fu-Kane vortex, the spin-resolved LDOS of vortex Majorana zero mode at the vortex core center is pinned to the direction that is parallel to the direction of the external magnetic field, as shown in Ref. [Phys. Rev. B 94, 224501 (2016)]. It is found that the spin-up LDOS $D_{\uparrow}(r=0)$ is nonzero while the spin-down component of LDOS $D_{\downarrow}(r=0)$ vanishes exactly at the vortex center, for a $+\hat{z}$ magnetic field. This gives rise to a dot-like D_{\uparrow} and a ring-like D_{\downarrow} . If we flip the field orientation, D_{\uparrow} features a ring-like structure while D_{\downarrow} shows a dot-like one. Again, Fu-Kane is a simplified model and the spin texture in Fu-Kane types of materials, e.g., $\text{FeTe}_{1-x}\text{Se}_x$, could be quantitatively and even qualitatively different.
- (ii) For a topological superconducting nanowire, the spin orientation of Majorana zero mode completely depends on the direction of the external magnetic field, as shown in Ref. [Phys. Rev. Lett. 112, 037001 (2014)]. We are not aware of any discussions on the spin-resolved spatial profile of MZM wavefunction along the radial direction of the nanowire. This could be an interesting question to theoretically explore in future works.

Q5: It is better to cite two more papers, they also focus on the emergent vortex topological invariant, but with different ideas. PRL 118, 147003 (2017); PRL 119, 047001 (2017).

Our reply: We thank the reviewer for pointing out these interesting references, which we have included in our revised manuscript.

Q6: A typo: The second to the last line of Fig.3 caption, (c) should be (e).

Our reply: We appreciate the reviewer for pointing out this typo. We have fixed it in the revised caption [note that it is now Fig. 4].

Reviewer #2

The paper presents a theoretical investigation of the band structure of vortex bound states in superconductors. It is demonstrated that the vortex bound states may be topologically non-trivial despite the topologically trivial nature of the bulk state. The results are illustrated by model calculations for HgTe-class materials using experimentally motivated parameters. It is proposed that lattice strain may be utilized to create or destroy Majorana zero modes in these materials.

Majorana quasiparticles in superconductors are widely investigated in the literature. The experimental observations of proposed Majorana zero modes are often accompanied by overly simplified theoretical models which only consider the topology of a single electronic band in most cases, leading to heated discussions concerning possible alternative explanations. The present manuscript makes an important step by highlighting the role of topologically trivial bands in the formation of topologically non-trivial vortex bound states, indicating that these bands cannot be neglected in a correct theoretical description. The symmetry considerations are supported by detailed numerical calculations using parameters for an experimentally studied class of materials, proposing a new research direction to observe Majorana zero modes in systems with a trivial band topology. Therefore, I am convinced that the manuscript will attract the interest of both experimental and theoretical groups working in the field and recommend it for publication in Nature Communications after the authors clarified a few remaining open questions.

Reply: We thank the reviewer for his/her recommendation of our manuscript. We are particularly grateful to the reviewer for his/her inspiring questions that motivate us to rethink deeply about our results and further significantly improve our manuscript. The major updates in our revised manuscript are summarized below:

1.) Vortex topological phase diagrams for Luttinger semimetal.

- To better illustrate the idea of trivial-band-induced vortex topology, we add a new figure [see Fig. 2] in the main text to systematically map out the vortex topological phase diagrams of Luttinger semimetal (LSM) as a function of chemical potential and lattice strain effects. While a tensile-strained LSM is a trivial band insulator, **electron and hole doping of this insulator will induce a topological nodal vortex phase and a Kitaev vortex phase**, respectively. We further illustrate the Majorana zero modes associated with these vortex phases by calculating

the surface local density of states (LDOS), as summarized in two new figures in Supplementary Note 2.5 [see Fig. 3 & 4]. These new updates unambiguously demonstrate the rise of Majorana physics from trivial electron bands.

2.) Topological vortex phases for the six-band Kane model.

- In the revised main text, we significantly update Fig. 4 (previously as Fig. 3) by including two topological vortex phase diagrams in Fig. 4 (b) and (c), both of which numerically prove the schematic phase diagram in Fig. 4 (a). We have also updated the spin-resolved surface LDOS for vortex Majorana zero modes two types of Kitaev vortices using the recursive Green's function method, as shown in Fig. 4 (d) and (e).

3.) Lattice model v.s. Continuum model

- In the supplementary materials, we have added a new section and a new figure [see Fig. 5 in Supplementary Note 5] to compare the results obtained from lattice and continuum models. Both modeling approaches agree with each other quantitatively.

Besides, at the end of this reply letter, we have appended detailed summaries of changes for both the main text and the supplementary information. With these revisions and updates, we have comprehensively addressed the comments and questions from both reviewers. We believe that our work is ready for immediate publication in Nature Communications.

Q1: In the manuscript I could not find a conclusive answer to the question whether Majorana bound states without nodal vortex states may be formed only from topologically trivial bulk bands. In Secs. II and III they appear simultaneously, meaning that the Majorana state at zero energy appears on top of the continuum of the nodal vortex band, and they could hybridize with each other if the rotational symmetry were broken. In Secs. IV and V, it appears that the main effect of the topologically trivial bands is to destroy the Majorana state created by the topologically non-trivial bands. The Majorana state may be restored by strain or symmetry breaking, but it still seems to require the presence of the topologically non-trivial bands in the six-band Kane model.

Although the manuscript is very carefully written and I could not find a statement that would indicate the presence of Majorana states without nodal vortex states in the topologically trivial regime of the bulk, a cursory look over the paper may arrive at the conclusion that Majorana states would be equally simple to realize in topologically trivial as in topologically non-trivial bulk bands.

Therefore, I think it would be necessary to clarify this question, for example by explaining whether any of the Majorana states sketched in Fig. 3(a) stem from an overall topologically trivial bulk band structure.

Reply: We thank the Reviewer for these important comments! To briefly summarize, two main questions, one at the conceptual level and another one at the material level, are raised by the Reviewer:

Q1a: Can Majorana zero mode (or equivalently Kitaev vortex) in principle arise from trivial bands?

Q1b: If so, which Majorana mode in the Kane-model phase diagram originates from trivial bands?

We will answer the above questions one by one in the following.

Reply to Q1a:

Yes, Kitaev vortex, without any coexisting nodal vortex, can indeed arise from trivial bands.

We agree with the Reviewer that the Kitaev vortex in the LSM model always coexists with the nodal vortex, as described by the process in Fig. 1 in the main text. The coexistence of the two phases likely originates from the fact LSM is a gapless critical phase. This motivates us to couple LSM with a lattice strain perturbation Σ_{str} in order to remove the quadratic band touching, namely, by carefully exploring the instability of topological vortices against Σ_{str} . In particular, while a compressive strain ($\Sigma_{str} < 0$) drives the system into a band-inverted Dirac semimetal, however, a tensile-strained ($\Sigma_{str} > 0$) LSM is a trivial band insulator with no band inversion.

The vortex topological phase diagram as a function of chemical potential μ and Σ_{str} is shown in Fig. R1 (a) below. We find that the coexisting Kitaev \oplus nodal phase is a universal feature for both LSM and the strain-induced Dirac semimetal phase, which agrees with our analysis in Fig. 1 in the main text. Moreover, a remarkable finding here is that with a tensile strain, **electron and hole doping of this trivial band insulator will lead to a nodal vortex and a Kitaev vortex**, respectively. This example thus provides an affirmative answer to **Q1a**.

In Fig. R1 (b), we further consider a rotational-symmetry-breaking strain effect Σ_{sb} and numerically map out a relevant vortex phase diagram. The strained system still features a two-fold rotation symmetry and thus admits a $Z_2 \times Z_2$ classification for vortex topology. In this case, a Kitaev vortex phase is symmetry labeled as Kitaev $_{\pm}$, where \pm denote the $C_2 = \pm 1$ sector that the vortex phase is living in. Remarkably, Fig. R1 (b) features all four topologically distinct vortex phases in this symmetry class, all of which are achieved when the normal-state band structure is a doped trivial band insulator.

Fig. R1 | Vortex topological phase diagrams (VTPD) of a strained LSM. (a) shows the VTPD as a function of Σ_{str} and μ . (b) shows the VTPD as a function of Σ_{sb} and μ , with a fixed $\Sigma_{str} = 0.3$ [white arrow in (a)]. The rotational symmetry breaking induced by Σ_{sb} updates the nodal vortex in (a) to the Kitaev₋ vortex in (b). Here \pm is used to represent the eigenvalue of the two-fold rotational symmetry. The figure is presented in Fig. 2 in the main text.

Fig. R2 | Surface LDOS for Kitaev_± vortex phases are shown in (a) and (b). The blue curve represents the LDOS $D_{tot}(\vec{r}_c, \omega)$ at the vortex core center $\vec{r}_c = (20,20)$ and the red curve is for the LDOS $D_{tot}(\vec{r}_b, \omega)$ at a position $\vec{r}_b = (30,30)$ far away from the vortex core. This figure is presented in Fig. 3 in Supplementary Note 2.5.

To provide direct evidence of how Majorana modes arise from these trivial-band induced Kitaev phases, we numerically calculate the surface local density of states (LDOS) by using the recursive Green's function technique to characterize the vortex Majorana zero modes. And the zero-bias peak signature of LDOS could directly indicate the existence of end Majorana zero modes in this system. The numerical results are shown in above Fig. R2 (a) and (b) for Kitaev₊ and Kitaev₋

vortex phases with a lattice size $L_x = L_y = 39$, where the blue curve represents the LDOS at the vortex core center ($x = y = 20$) and the red curve is for the LDOS at a position ($x = y = 30$) far away from the vortex core. Both cases clearly show the Majorana-induced zero-bias peak signature at the vortex core center.

We have included these new figures and results in the revised manuscript.

Reply to Q1b:

The nodal-originated Kitaev vortex, now labeled as Kitaev_ phase, is an outcome of the trivial bands in the Kane model.

Unlike the Luttinger semimetal, the trivial bands and topological bands in the 6-band Kane model are strongly coupled to each other and it is impossible to thoroughly get rid of the topological bands. Nonetheless, as shown in Fig. 3 (a) [i.e., the previous Fig. 2(a)], if the trivial bands are turned off, the remaining four bands resemble a standard topological insulator Hamiltonian, which only supports the Kitaev vortex phase. As a result, the emergence of the nodal vortex is purely a trivial-band effect. Consequently, breaking the protecting symmetry of the nodal vortex will lead to a trivial-band-induced Kitaev vortex, i.e., the Kitaev_ phase in Fig. 4.

While the original Fig. 3 (a) [also see Fig. R3 (a) below] is schematically plotted using symmetry argument, we have numerically verified its validity by mapping out two new vortex topological phase diagrams, one as a function of μ and Σ_{sb} for a fixed Σ_{str} [see Fig. R3 (b)] and another as a function of μ and Σ_{str} for a fixed Σ_{sb} [see Fig. R5 (c) in our reply to your Q3 below]. Here Σ_{str} and Σ_{sb} are lattice strain parameters defined similarly to those for the Luttinger semimetal in Fig. R1. We strongly believe that these new vortex phase diagrams have further solidified our conclusion of trivial-band induced Kitaev vortex phase in the Kane-model systems.

In addition, we further calculate the surface LDOS using an iterative Green's function method for both Kitaev vortex phases, as shown in Fig. R5 (d) – (i) [see our reply to your Q3 below]. The zero-bias peak signals clearly provide direct evidence of Majorana bound states in both vortex phases, irrespective of whether the Kitaev vortex is trivial-band originated or not.

In the revised manuscript, Fig. R3 and Fig. R5 are combined to form the Fig. 4 in the main text. We believe these new results have further established the central role of trivial bands in the vortex physics of HgTe-class materials.

Fig. R3 | Strain-controlled Majorana engineering of HgTe-class materials. (a) schematically shows the evolution of the local density of state (LDOS) at the vortex core center as a function of bias voltage by tuning both the in-plane lattice strain strength Σ_{str} and the C_3 -symmetry breaking perturbation Σ_{sb} . The Kitaev-trivial-Kitaev transition in (a) is numerically verified by both mapping the Σ_{str} - Σ_{sb} VTPD in (b) at a fixed $\mu = 0.1$ eV. This figure is presented in Fig. 4 in the main text.

Q2: Generally, the eigenstates should be classified according to the irreducible representations of the group. For the point groups C_n these may indeed be described by the angular momentum quantum numbers $J_z=0, \dots, n-1$, as shown in Table I. However, including additional mirror symmetries, for example in the C_{nv} groups, will combine some angular momentum quantum numbers into a multi-dimensional irreducible representation, inducing degeneracies in the high-symmetry points. Would these additional symmetries influence the conclusions about the emergence of Kitaev and nodal vortex states? This could be relevant since the point groups O_h and T_d mentioned for the investigated material class in the first paragraph of Sec. III also contain mirrorings. The model considered in Sec. III has a continuous rotational symmetry around the z axis, which is broken in real systems by the crystal structure. Do the conclusions change if such symmetry-breaking terms are taken into account? In summary, I think it would be beneficial to discuss in more detail which symmetries are necessary and which symmetries are not allowed for the conclusions to hold.

Reply: These are both excellent questions!

For the **first question** regarding the mirror symmetries, it is true that C_{nv} double group usually feature 2D irreducible representations, which dramatically changes the band configurations for bulk bands. However, when a magnetic field is applied to generate a vortex, **all in-plane mirror symmetries parallel to the magnetic field (i.e., vortex-line orientation) are broken by the superconducting winding phase**. Limiting ourselves to point groups, the only remaining crystal

symmetries are the out-of-plane mirror, inversion symmetry, and roto-inversion symmetries, none of which can induce 2D or higher-dimensional irreps for CdGM states along k_z . This is why we think our classification table should be complete for point group symmetries. Of course, generalization to space group symmetries could further enrich our classification table, which is beyond the scope of this work.

Back to the Luttinger semimetal or Kane model, even though its bulk normal state features a high-symmetric O_h point group, the only important symmetries for the vortex-line Hamiltonian belong to a reduced subgroup of O_h , e.g., C_3 group. Therefore, our classification table is directly applicable to understanding the vortex topology of our target systems.

The **second question** can be rephrased as follows:

Do the main results of this work depend on the modeling approach, e.g., continuum model and lattice model?

Motivated by this question, we have systematically compared the vortex topological phase diagrams obtained by continuum models and lattice-regularized models for both the 4-band Luttinger model and the 6-band Kane model. As shown in the following Fig. R4, **both modeling approaches agree quantitatively well with each other. We, therefore, conclude that the main results in this work are robust and independent of the modeling approaches.**

This good agreement between different modeling methods can be understood as follows. For a $k \cdot p$ continuum model, the vortex Hamiltonian is decomposed into different subspaces labeled by total angular momentum J_z , namely, $H_{BdG} = \sum_{J_z} [\oplus H_{J_z}^{kp}]$. In particular for our systems, the $H_{J_z=0}^{kp}$ subspace leads to a Kitaev vortex while the $H_{J_z=1}^{kp} \oplus H_{J_z=-1}^{kp}$ subspaces give rise a nodal vortex. When the full rotational symmetry is broken, for example, $C_\infty \rightarrow C_3$ for a tight-binding model. The angular momentum sectors that used to contribute to Kitaev and nodal vortices are still well-defined in a reduced C_3 symmetry, but there now exists an infinite number of equivalent angular momentum sectors. For example, $H_{J_z=0}^{TB}$ of the tight-binding model is effectively a superposition of $H_{J_z=0}^{kp} \oplus H_{J_z=\pm 3}^{kp} \oplus H_{J_z=\pm 6}^{kp} \oplus \dots$ with off-diagonal coupling matrices. If the band structure of $k \cdot p$ model fit that of tight-binding model very well (as in our case), the off-diagonal coupling terms are expected to be small so that the essential vortex physics in $J_z = 0$ will not be significantly impacted by the couplings to higher angular momentum sectors. This is why results from our continuum model agree quantitatively well with those from the tight-binding model.

Fig. R4 | Continuum models v.s. lattice models. (a) and (d) are the bulk band structures, where the green shaded region denotes the energy range where continuum and lattice models fit well with each other. Clearly, within these energy windows, the VTPDs shown in (b) and (c), as well as in (e) and (f), also agree well. This figure is presented in Fig. 5 in Supplementary Note 5.

Q3: The breaking of the rotational symmetry in Sec. VI, Fig. 3 and Supplementary Fig. 3 is rather vaguely explained. It would be useful to add what was exactly the symmetry-breaking term and which parameter values were used in the calculations illustrated by the spin-resolved LDOS of the zero-energy states. I think it would also be useful to shortly clarify what the spin-up and spin-down components in the Γ wave function mean, since the bulk Hamiltonian is invariant under time reversal where spin-up and spin-down states should be equivalent. Are the spin-up and spin-down components connected to the sign of the phase winding in the pairing function around the vortex?

Reply: We are grateful to the reviewer for raising these important questions!

Motivated by this question, we reexamined our previous approach to extract spin-resolved LDOS and realized its potential limitations. To overcome these limitations, we develop a recursive Green's function algorithm that can accurately calculate the spin-resolved surface LDOS for semi-infinite slab geometry. With this new algorithm, we recalculated the spin-resolved LDOS plots for

vortex MZMs and updated the relevant plots in the main text. In the following, we will briefly describe our previous method and discuss its simplifications and limitations. We will then move on to describe the Green's function approach and discuss the updated results we obtained.

Our previous approach is based on the fact that **whenever a vortex topological phase transition occurs, the surface vortex Majorana modes feature a diverging localization length $\xi_M \rightarrow \infty$ and thus become gapless vortex-line bulk states at $k_z = 0$** . Therefore, by simply reading out this critical bulk-state wavefunction, we arrive at an effective surface Majorana wavefunction in the delocalized limit. By projecting the Majorana wavefunction onto different spin sectors, we thus arrive at the spin-resolved LDOS plots shown in Fig. 3 of our previous version of the manuscript. However, being an approximation method, the **pros** and **cons** of this approach are both quite obvious:

Pros: This method is of great computational efficiency, since it only involves 2D calculations instead of 3D ones.

Cons: This method is valid when ξ_M diverges. However, it is unclear how accurate this approach would be, when ξ_M is finite or small.

To overcome these issues, we formulate a recursive Green's function algorithm that enables us to calculate the spin-resolved surface local density of states for any location in the phase diagram with both lattice strain effect Σ_{str} and rotational symmetry-breaking term Σ_{sb} . Unlike the previous approach, the Green's function method requires a lattice-regularized tight-binding model. Our computation capability allows us to consider an in-plane 35×35 lattice and place the vortex core at the center of our lattice. The z-direction of our system is essentially semi-infinite due to the recursive nature of the Green's function algorithm. The rotation symmetry breaking that leads to nodal vortex \rightarrow Kitaev vortex transition is the Σ_{sb} term, the same lattice strain perturbation occurring in the vortex phase diagram in Fig. R3. We present this term and the parameters used for all calculations clearly in the revised manuscript.

Fig. R5 shows our recalculated spin-resolved LDOS D_\uparrow and D_\downarrow for two types of Kitaev vortices. To be comparable with experimental LDOS probes, **the LDOS in our plots is showing only the electron part of the Majorana wavefunctions**. We note that the ring-like pattern in our earlier results is absent in the updated calculation. Nonetheless, **the two Kitaev $_{\pm}$ phases still display distinct spin textures, where $D_\uparrow(r=0) > D_\downarrow(r=0)$ for the Kitaev $_-$ MZM and $D_\uparrow(r=0) < D_\downarrow(r=0)$ for the Kitaev $_+$ MZM**. In experiments, this distinct spin pattern should still be visible with the help of state-of-the-art spin-polarized scanning tunneling microscopy.

We note that it is still possible that the ring-like pattern indeed exists in the LDOS calculations of Fig. 5, but they are simply invisible due to the spatial resolution of our simulation. For example, if the radius of the ring is smaller than the lattice constant of the model, our Green's function approach is incapable of visualizing such a “sub-lattice” spatial feature.

Fig. R5 | Strain-controlled Majorana engineering of HgTe-class materials. (c) shows the Kitaev-trivial-Kitaev transition is numerically verified by mapping the Σ_{str} - Σ_{sb} VTPD at a fixed $\Sigma_{sb} = 0.2$ eV. The spatial spin-resolved surface LDOSs at a zero-bias voltage is numerically calculated for the Kitaev₊ vortex in (d) - (f) and the Kitaev₋ vortex in (g) - (i), respectively. These two topologically equivalent Kitaev vortex phases can be clearly distinguished by their distinct zero-bias spin textures as shown in (f) with $D_{\uparrow} < D_{\downarrow}$ and (i) with $D_{\uparrow} > D_{\downarrow}$ at the vortex core center $\vec{r}_c = (18, 18)$. This figure is presented in Fig. 4 in the main text.

We also hope to clarify that while the normal-state Hamiltonian respects time-reversal symmetry (TRS), the BdG one with a vortex does not. Therefore, the spin-up and spin-down LDOS can be, in principle, different as a result of the TRS breaking, as found in Fig. R5. In fact, this imbalance of LDOS between different spin sectors is quite common for vortex systems. For example, a Fu-Kane vortex in a doped topological insulator also features a similar phenomenon and an example of this can be found in Ref. [Phys. Rev. B 94, 224501 (2016)].

Finally, we agree with the reviewer that the sign of phase winding will control the spin texture of vortex MZM. This can be intuitively understood by noting that the vortex Hamiltonian $H_{BdG}(r, \phi)$ with the superconducting winding phase $\Delta_0(r)e^{i\phi}$ is transformed into $H_{BdG}(r, -\phi)$ under time-reversal symmetry. Therefore, $D_{\uparrow}(r)$ and $D_{\downarrow}(r)$ for $H_{BdG}(r, \phi)$ must equal to the $D_{\downarrow}(r)$ and $D_{\uparrow}(r)$ for $H_{BdG}(r, -\phi)$, respectively, as long as the vortex-free Hamiltonian $H_{BdG}(r, 0)$ preserves TRS.

Other minor comments:

Q4: In the first paragraph on page 4, "positive $m_{\{1\}}$ " and "negative $m_{\{2\}}$ " are mentioned, which contradicts the signs shown in Fig. 1.

Our reply: We thank the reviewer for pointing out this typo. We have fixed this typo to make everything self-consistent.

Q5: In the Hamiltonian defined in Eqs. (20) and (21) of the Supplementary Material, the pairing potential seems to connect spin-up electrons with spin-up holes, although it describes s-wave pairing. Does the spin-up hole denote the annihilation of a spin-down electron? In the text, it is only highlighted that the e and h indices describe opposite wave vectors.

Reply: Yes, our notation in Eq. (20) is defined as follows:

$$\Xi |J_z, s\rangle_e \rightarrow |-J_z, -s\rangle_h$$

where Ξ is the particle-hole symmetry and $s = \uparrow, \downarrow$ denotes the spin. For example, the pairing term between $|\frac{3}{2} \uparrow\rangle_e$ and $|\frac{3}{2} \uparrow\rangle_h$ describes a zero-angular-momentum spin-singlet s-wave Cooper pairing between two $|\frac{3}{2} \uparrow\rangle_e$ electrons. We have revised our supplementary to clarify our notation.

Q6: In Eq. (21) in the Supplementary Material, the connection of the v and \tilde{v} parameters to the parameters in the Hamiltonian in Eq. (16) seems not be explained.

Reply: We thank the reviewer for pointing out this. We have added the definition of these parameters in the revised supplementary material.

Q7: In the line above Eq. (31) on page 6 of the Supplementary Material, $k_{\{x\}}+k_{\{y\}}$ should probably be replaced by $k_{\{x\}}+ik_{\{y\}}$.

Reply: We appreciate the reviewer for pointing out this typo. We have revised the manuscript accordingly.

Q8: In the first paragraph of Supplementary Sec. IID on page 3, the $m_{\{1\}}$, $m_{\{2\}}$ and $v_{\{z\}}$ parameters used for the simulation appear to be missing.

Our reply: We thank the reviewer for pointing out this. We have revised our manuscript to make sure all parameters used in the simulations are provided in either the main text or the supplementary material.

A summary of changes in the main text (marked in blue words)

1. In the abstract, we revise one sentence “A feasible scheme of strain-controlled Majorana engineering and **experimental signatures for Majorana modes** are also discussed.”.
2. In paragraph 2 of the introduction, we add one sentence to mention our new results for Luttinger semimetal: “**Furthermore, a tensile-strained LSM is found to be a bulk-trivial yet vortex-exotic band insulator, which harbors nodal and Kitaev vortex phases in the presence of electron and hole dopings, respectively.**”
3. On page 2, in paragraph 3 of the introduction, we revise one sentence to mention our new results for the surface local density of state simulation: “**Experimental signatures of the proposed vortex topological physics are discussed in the details.**”.
4. In both paragraph 2 and the caption of Table (I) of Sec. II, we clarify that Table. (I) works for **a general s-wave spin-singlet superconductor**.
5. On page 4, we add a new figure [Fig. 2] to demonstrate the strain-tuned topological vortices in a doped trivial band insulator.
6. On pages 4 & 5, we add three new paragraphs to describe the new results in Fig. 2.
7. On page 6, we revise **the first paragraph in Sec. V** to briefly discuss our motivation to study the strain effect.
8. On pages 6 & 7, we revise two sentences: “**Similar to the scenario of LSM**, a compressive” and “instead of spoiling it, **which agrees with our LSM-based VTPD in Fig. 2.**”.
9. On pages 7 & 8, we rephrase all the related discussions on the experimental signatures of strain controlled Majorana zero modes in realistic materials based on new results.
10. On page 8, we revise Fig. 4 by adding numerically calculated phase diagrams and surface local density of states.

A summary of changes in the supplementary information (marked in blue words)

1. On page 5, we clarify the notation of the basis for the BdG Hamiltonian in Eq. (20).
2. On page 5, we add the definition of \tilde{v} for Eq. (22).
3. On page 6, we correct the definition of k_{\pm} for Eq. (32).
4. On page 7, in the supplementary note 2.4, we revise the description for Fig. 2.
5. On pages 7 - 9, we add a new supplementary note 2.5 to discuss our method to calculate the surface local density of states and the results for Luttinger semimetals, with two new figures.

6. On page 13, in supplementary note 4, we remove our previous method to calculate the surface local density of states. It has been replaced by the new Green's function method in supplementary note 2.5.
7. On pages 13 & 14, we add a new supplementary note 5 with a new figure to discuss the similar results of the tight-binding model and continuum model.

REVIEWERS' COMMENTS

Reviewer #1 (Remarks to the Author):

I would like to thank the authors for their careful consideration of my questions and the adjustment to the manuscript. I believe that my concerns were in principle addressed, providing substantial additional simulations in their detailed reply.

I am now happy to recommend to publish this manuscript on Nature Communications.

Reviewer #2 (Remarks to the Author):

The authors convincingly replied to the comments of the Reviewers, and considerably extended the discussion by the results of further calculations. In my opinion, these revisions helped in communicating and supporting the main result of the manuscript more clearly, namely that Majorana zero modes may be formed in vortices even if the underlying bulk electronic structure is topologically trivial. Based on the expected large impact of the findings and the high quality of the manuscript, I recommend it for publication in Nature Communications.

I have a few minor comments which the author may consider before publication:

1. In the new figures displaying the tight-binding model calculations (Fig. 2, Fig. 4(b),(d), Supplementary Fig. 5(b),(e)), it does not seem to be mentioned what the colour coding denotes.

2. It is stated in the caption of Fig. 2 that "The model parameters for both calculations are the same as those in Fig. 1." However, the parameters are gradually turned on between the panels of Fig. 1, so it should be clarified which panel this sentence refers to.

Furthermore, it may be useful to add to Fig. 2(a) for which values of the strain the bulk is topologically non-trivial or topologically trivial, to emphasize the point of the manuscript. This is explained in the text, but apparently not in the figure or the caption.

3. The references in the main text to the Supplementary Notes should be checked, since at some points they contain Roman numerals while in the Supplementary Material the numbers are Arabic and may also contain subsections.

4. In the third paragraph on page 4, the strain is described by the term $-\Sigma_{\text{str}}\gamma_1$ in the Hamiltonian. Based on the representation of the strain in Sec. V and the Supplementary Material (Eq. (18)), the gamma matrix here is probably not correct, it may be γ_5 instead.

In the same paragraph and the next one, the inversion symmetry is represented by the matrix γ_0 . Since γ_0 was defined as the identity matrix in the first paragraph on page 3, this does not appear to be correct.

5. In Fig. 3(d), the orange and green lines denote the same angular momentum sectors as in Fig. 1, but the positive and negative curvatures are exchanged. This may be intentional since the two figures describe different models, but it could be checked since there is an analogy between the models.

6. In the caption of Fig. 4, it is stated that panels (d)-(f) show the Kitaev_{+} vortex and panels (g)-(i) show the Kitaev_{-} vortex, in agreement with the reply to the Reviewers. However, in the text the Kitaev_{+} and Kitaev_{-} vortices are interchanged compared to the caption, and this convention seems to agree with the results in Supplementary Fig. 4 referenced here. This ambiguity should be clarified, preferably by explaining in the figure or the caption which panels belong to the

white dot and the white square in Fig. 4(c), respectively.

7. Equation (29) in the Supplementary Material only contains 7 elements, although it should be an 8-dimensional vector.

8. Since the surface Green's function method in Supplementary Note 2.5 is referred to as a standard technique, it would be useful to add a reference, e.g. M. P. Lopez Sancho et al., *J. Phys. F: Met. Phys.* 15, 851 (1985).

9. In Supplementary Note 4, the number of the Section referenced in the second sentence is missing.

10. In Supplementary Fig. 5 and the corresponding discussion, it is not stated explicitly which of panels (b), (c), (e) and (f) correspond to the tight-binding model and which to the continuum model.

Reviewer #1 (Remarks to the Author):

I would like to thank the authors for their careful consideration of my questions and the adjustment to the manuscript. I believe that my concerns were in principle addressed, providing substantial additional simulations in their detailed reply.

I am now happy to recommend to publish this manuscript on Nature Communications.

Reply: We thank the Reviewer #1 for his/her recommendation.

Reviewer #2 (Remarks to the Author):

The authors convincingly replied to the comments of the Reviewers, and considerably extended the discussion by the results of further calculations. In my opinion, these revisions helped in communicating and supporting the main result of the manuscript more clearly, namely that Majorana zero modes may be formed in vortices even if the underlying bulk electronic structure is topologically trivial. Based on the expected large impact of the findings and the high quality of the manuscript, I recommend it for publication in Nature Communications.

Reply: We thank the Reviewer #2 for his/her recommendation.

I have a few minor comments which the author may consider before publication:

1. In the new figures displaying the tight-binding model calculations (Fig. 2, Fig. 4(b),(d), Supplementary Fig. 5(b),(e)), it does not seem to be mentioned what the colour coding denotes.

Reply: We thank the reviewer for the suggestion. In the revised manuscript,

(i) We add “Both VTPDs are mapped out by calculating the vortex-state energy gap at $k_z=0$, whose logarithmic value is shown by the colors in (a) and (b). Vortex topology changes whenever the vortex-state gap closes.” for the caption of Fig. 2.

(ii) We add “The colors in (b) and (c) represent the logarithmic value of the vortex energy gap at $k_z = 0$.” for the caption of Fig. 4 (b) and (c). We also revise “Besides, the color map plots of the spatial spin-resolved surface LDOS...” for the caption of Fig. 4 (d), (e), (g) and (h).

In the revised supplementary material,

(i) We revise “The color map plots of the spin-resolved surface LDOS...” for the caption of Fig. 4.

(ii) We add “For the results based on tight-binding model, the VTPDs are achieved by calculating the vortex energy gap at $k_z = 0$, whose logarithmic value is shown by the colors in (b) and (e).” for the caption of Fig. 5 (b) and (e).

2. It is stated in the caption of Fig. 2 that “The model parameters for both calculations are the same as those in Fig. 1.” However, the parameters are gradually turned on between the panels of Fig. 1, so it should be clarified which panel this sentence refers to.

Furthermore, it may be useful to add to Fig. 2(a) for which values of the strain the bulk is topologically non-trivial or topologically trivial, to emphasize the point of the manuscript. This is explained in the text, but apparently not in the figure or the caption.

Reply: We thank the reviewer for the suggestion. We revise “Fig. 1” to “Fig. 1 (f)” for the caption. Following the Reviewer’s suggestion, we emphasize which parameter region is for a trivial insulator region by adding “Specifically, the normal state is a topologically trivial insulator for $\Sigma_{str} > 0$ and a Dirac semimetal for $\Sigma_{str} < 0$.” for the caption of Fig. 2 (a).

3. The references in the main text to the Supplementary Notes should be checked, since at some points they contain Roman numerals while in the Supplementary Material the numbers are Arabic and may also contain subsections.

Reply: We have double checked each reference in the main text to the supplementary note and unify all the references in Arabic, following the format requirement of Nature Communications.

4. In the third paragraph on page 4, the strain is described by the term $-\Sigma_{str}\gamma_{1}$ in the Hamiltonian. Based on the representation of the strain in Sec. V and the Supplementary Material (Eq. (18)), the gamma matrix here is probably not correct, it may be γ_{5} instead.

In the same paragraph and the next one, the inversion symmetry is represented by the matrix γ_{0} . Since γ_{0} was defined as the identity matrix in the first paragraph on page 3, this does not appear to be correct.

Reply: We thank the reviewer for pointing out this typo! Yes, it should be γ_5 for the Σ_{str} term. The matrix representation of the inversion operator $P = \gamma_0$ is correctly defined for the normal-state Hamiltonian of the Luttinger semimetal model. As we discuss in the second paragraph of the Section “Vortex Topology From Trivial Bulk Bands”, the basis of the Hamiltonian is consisting of

the p_x, p_y orbitals and spin, $|\Psi_{\Gamma_8}\rangle = (|p_+, \uparrow\rangle, |p_+, \downarrow\rangle, |p_-, \uparrow\rangle, |p_-, \downarrow\rangle)^T$. Each component of $|\Psi_{\Gamma_8}\rangle$ has even parity under inversion. However, we guess the Reviewer #2 is asking this question because he/she may be confused with the inversion for the normal-state Hamiltonian and the superconducting BdG Hamiltonian. Therefore, in the revised manuscript, we clarify this point by emphasizing that the representation of inversion operator $P = \gamma_0$ is for the normal-state Hamiltonian.

5. In Fig. 3(d), the orange and green lines denote the same angular momentum sectors as in Fig. 1, but the positive and negative curvatures are exchanged. This may be intentional since the two figures describe different models, but it could be checked since there is an analogy between the models.

Reply: We thank Reviewer #2 for bringing out this interesting question!

Indeed, the LSM model and the Kane model are deeply connected to each other. Starting from the six-band Kane model, one can project out the Γ_6 bands and focus on the Γ_8 -band manifold, which immediately leads us to the Luttinger model. Therefore, the Kane model will inherit the vortex topology induced by the LSM bands, as we have emphasized in the main text. Nonetheless, the angular momentum information of vortex states is a beyond-topology feature, which will sensitively depend on the specific choice of model parameters.

To see this, we carry out this band downfolding procedure in Supplementary Note 2.1 for the HgTe model, to get an effective LSM model. We find that the HgTe-based LSM model exactly reproduces the LSM model in Fig. 1 & Fig. 2, but their model parameters are having opposite signs. Therefore, **it is exactly this sign difference in the model parameters that leads to the subtle difference in the angular momentum values of vortex-bound states for Fig. 1 (f) and Fig. 3 (d)**. For example, if we flip the sign of parameters in the LSM model to follow the convention in the HgTe model, we will get the same angular momentum label for the nodal vortex spectra in Fig. 1 (f) and Fig. 3 (d). Again, the key conclusion for the vortex topology of LSM & HgTe models is not influenced by these different parameter choices.

To clarify this subtle issue, we add a detailed comparison between the parameters of the LSM model and those for the projected LSM model from the Kane model in Supplementary Note 2.1. We have also included one sentence “**Note that $Q_1 = 1$ in Fig. 1 is due to a different parameter choice in the LSM model, which we elaborate in the Supplementary Note 2.1.**” in the revised manuscript.

6. In the caption of Fig. 4, it is stated that panels (d)-(f) show the Kitaev_{+} vortex and panels (g)-(i) show the Kitaev_{-} vortex, in agreement with the reply to the Reviewers. However, in the text the Kitaev_{+} and Kitaev_{-} vortices are interchanged compared to the caption, and this convention seems to agree with the results in Supplementary Fig. 4 referenced here. This ambiguity should be clarified, preferably by explaining in the figure or the caption which panels belong to the white dot and the white square in Fig. 4(c), respectively.

Reply: We thank the reviewer for pointing out this typo. We correct the notations in the caption of Fig. 4.

7. Equation (29) in the Supplementary Material only contains 7 elements, although it should be an 8-dimensional vector.

Reply: We thank the reviewer for pointing out this typo. It has been corrected.

8. Since the surface Green's function method in Supplementary Note 2.5 is referred to as a standard technique, it would be useful to add a reference, e.g. M. P. Lopez Sancho et al., J. Phys. F: Met. Phys. 15, 851 (1985).

Reply: We thank the reviewer for pointing out this missing reference, which we have included in the revised manuscript.

9. In Supplementary Note 4, the number of the Section referenced in the second sentence is missing.

Reply: We thank the reviewer for pointing out this typo. It has been corrected.

10. In Supplementary Fig. 5 and the corresponding discussion, it is not stated explicitly which of panels (b), (c), (e) and (f) correspond to the tight-binding model and which to the continuum model.

Reply: We thank the reviewer for the suggestion. We have added one sentence “Here (b) and (e) are obtained from the tight-binding models, while (c) and (f) are based on the continuum models.” in the caption of Fig. 5.